# Physical descriptor for the Gibbs energy of inorganic crystalline solids and temperature-dependent materials chemistry

Christopher J. Bartel[1], Samantha L. Millican[1], Ann M. Deml[2,3], John R. Rumptz[1], William Tumas[3], Alan W. Weimer[1], Stephan Lany [3], Vladan Stevanović[2,3], Charles B. Musgrave[1,3,4] & Aaron M. Holder [1,3]

The Gibbs energy, $G$, determines the equilibrium conditions of chemical reactions and materials stability. Despite this fundamental and ubiquitous role, $G$ has been tabulated for only a small fraction of known inorganic compounds, impeding a comprehensive perspective on the effects of temperature and composition on materials stability and synthesizability. Here, we use the SISSO (sure independence screening and sparsifying operator) approach to identify a simple and accurate descriptor to predict $G$ for stoichiometric inorganic compounds with ~50 meV atom$^{-1}$ (~1 kcal mol$^{-1}$) resolution, and with minimal computational cost, for temperatures ranging from 300–1800 K. We then apply this descriptor to ~30,000 known materials curated from the Inorganic Crystal Structure Database (ICSD). Using the resulting predicted thermochemical data, we generate thousands of temperature-dependent phase diagrams to provide insights into the effects of temperature and composition on materials synthesizability and stability and to establish the temperature-dependent scale of metastability for inorganic compounds.

[1] Department of Chemical and Biological Engineering, University of Colorado, Boulder, CO 80309, USA. [2] Department of Metallurgical and Materials Engineering, Colorado School of Mines, Golden, CO 80401, USA. [3] National Renewable Energy Laboratory, Golden, CO 80401, USA. [4] Department of Chemistry and Biochemistry, University of Colorado, Boulder, CO 80309, USA. Correspondence and requests for materials should be addressed to C.B.M. (email: charles.musgrave@colorado.edu) or to A.M.H. (email: aaron.holder@colorado.edu)

The progression of technology throughout history has been preceded by the discovery and development of new materials[1]. While the number of possible materials and the variety of their properties is virtually limitless, discovery of new compounds with superior properties that are also stable (or persistently metastable) and synthesizable is a tremendous undertaking that remains as an ongoing challenge to the materials science community[2–5]. The leading paradigm in this effort is the use of first-principles computational methods, such as density functional theory (DFT), and materials informatics to rapidly populate, augment and analyze computational materials databases and screen candidate materials for target properties[6,7]. However, despite the exploding growth of these databases with the number of compiled entries currently exceeding 50 million[8], only a small fraction of realized or potential materials have known Gibbs energies of formation, $\Delta G_f(T)$, which is critical for predicting the synthesizability and stability of materials at conditions of interest for numerous applications which operate at elevated temperature including thermoelectrics[9], ceramic fuel cells[10], solar thermochemical redox processes[11], and $CO_2$ capture[12].

Experimental approaches for obtaining $\Delta G_f(T)$ are demanding, and the number of researchers using calorimetry to determine $\Delta G_f(T)$ is significantly smaller than those focused on the discovery and synthesis of new materials. Ab initio computational approaches for determining $\Delta G_f(T)$, which involve calculating the vibrational contribution to $G(T)$ as a function of volume[13], have benefited from recent advances that reduce their computational cost[14,15]. However, despite these advances, calculating the vibrational entropy of phonons quantum mechanically is still computationally demanding, with computed $G(T)$ available for fewer than 200 compounds in the Phonon database at Kyoto University (PhononDB)[16]. Highly populated and widely used materials databases currently tabulate 0 or 298 K enthalpies of formation, $\Delta H_f$, which neglect the effects of temperature and entropy on stability. As a result, the growth of computational materials databases has far outpaced the tabulation of measured or computed $\Delta G_f(T)$ of materials, precluding researchers from obtaining a comprehensive understanding of the stability of inorganic compounds.

The use of machine learning and data analytics to accelerate materials design and discovery through descriptor-based property prediction is becoming a standard approach in materials science[17–24], however, these techniques have not previously been used to predict the Gibbs energies of inorganic crystalline solids. Techniques based on symbolic regression have also shown that

fundamental physics can be algorithmically obtained from experimental and computed data in the form of optimized analytical expressions of intrinsic properties (features)[25–27]. In this work, we apply a recently developed statistical learning approach, SISSO (sure independence screening and sparsifying operator)[28], to search a massive (~$10^{10}$) space of mathematical expressions and identify a descriptor for experimentally obtained $G(T)$ that for the first time enables $\Delta G_f(T)$ to be readily obtained from high-throughput DFT calculations of a single structure (i.e., a single unit cell volume). The descriptor is identified using experimental data[29] for 262 solid compounds and tested on a randomly chosen excluded set of 47 compounds with measured $G(T)$ and 131 compounds with first-principles computed[16] $G(T)$. We then apply this descriptor to ~30,000 unique crystalline solids tabulated in the Inorganic Crystal Structure Database (ICSD) to generate the most comprehensive thermochemical data of inorganic materials to date.

## Results

**Trends in the Gibbs energies of compounds and elements.** Despite the variations of composition and structure exhibited by different inorganic crystalline compounds, $G(T)$ behaves remarkably similarly over a wide range of materials (Fig. 1a). This similarity prompts the hypothesis that although the underlying physical phenomena that give rise to $G(T)$ are complex to describe individually, a physically motivated descriptor could be predictive. The origin of the similar behavior of $G(T)$ can be understood from well known thermodynamic relations, specifically that $\left(\frac{\partial G}{\partial T}\right)_P = -S \leq 0$ for mechanically stable compounds and that $G(T)$ must have negative concavity: $\left(\frac{\partial^2 G}{\partial T^2}\right)_P = -\left(\frac{\partial S}{\partial T}\right)_P = -\frac{C_P}{T} \leq 0$. Indeed, the negative first and second derivatives of experimental Gibbs energies as a function of temperature persist across the composition space of a diverse set of mechanically stable stoichiometric solid compounds (Fig. 1a). We reference the Gibbs energy, $G$, with respect to the formation enthalpy at 298 K, $\Delta H_f$, because $\Delta H_f$ is readily obtained using existing high-throughput computational methods—DFT total energy calculations and a suitable correction for the elemental phases[30–33]:

$$G^\delta(T) = G(T) - \Delta H_f(298\,\text{K}) \qquad (1)$$

As expected, the temperature- and material-dependence of the enthalpic contribution to the Gibbs energy, $G^\delta$, is small relative to the entropic contribution ($TS$). If the standard state formation

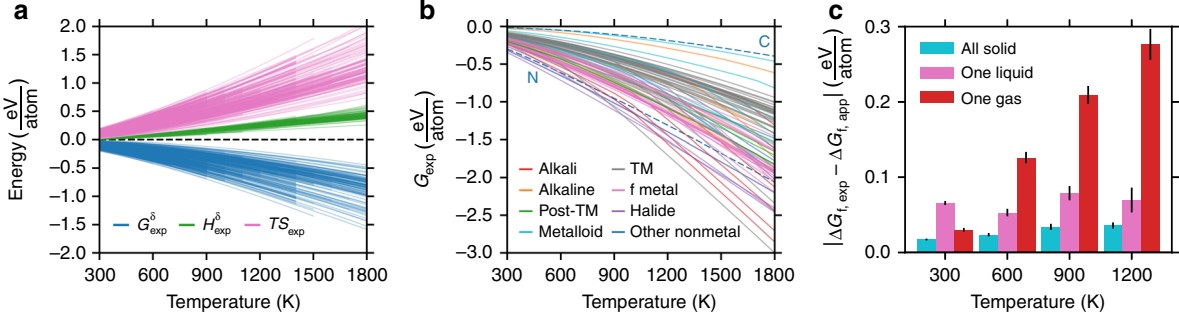

**Fig. 1** Contributions to the Gibbs energies of compounds. **a** Experimentally obtained thermodynamic functions of 309 inorganic crystalline solid compounds obtained from FactSage. $G^\delta$ is defined in Eq. (1). $H^\delta$ is the temperature dependence of the enthalpy normalized to be zero at 298 K (Supplementary Eq. 1), $S$ is the absolute entropy, and $T$ is temperature. The subscript, exp, indicates the quantity is obtained from experimental data. **b** Experimentally determined absolute Gibbs energies of 83 elements obtained from FactSage. $G_C$ ("C") and $G_N$ ("N") are dashed and labeled as they are mentioned in the text. The subscript, exp, indicates the quantity is obtained from experimental data. **c** Mean absolute error in assuming a cancellation of solid vibrational entropy between the compound and the elements comprising it. $\Delta G_f(T)$ is defined in Eq. (3). The subscript, app, stands for approximation and $\Delta G_{f,app}(T)$ is defined in Eq. (4). The error bars are standard errors of the sample mean. A violin plot corresponding with each bar is provided in Supplementary Fig. 1

enthalpy, $\Delta H_f$, is known, the temperature dependence of the enthalpy is reliably predicted with a simple linear fit ($R^2 \sim 0.97$, Supplementary Eq. 1) for the 309 solid compounds considered in this work. This is assumed implicitly when the quasiharmonic approximation[34] of the phonon free energy is used to obtain $G(T)$, but is quantified here across a broad composition and temperature space.

In addition to the thermodynamic quantities $\Delta H_f$ and $G^\delta(T)$, the chemical potentials of the elements, $G_i(T)$, also have a critical role in the Gibbs formation energy, $\Delta G_f(T)$, and thus the temperature-dependent stability of a given compound:

$$\Delta G_f(T) = \Delta H_f(298\,K) + G^\delta(T) - \sum_{i=1}^{N} \alpha_i G_i(T) \qquad (2)$$

where $N$ is the number of elements in the compound, $\alpha_i$ is the stoichiometric weight of element $i$ and $G_i$ is the absolute Gibbs energy of element $i$. While even at low temperatures the differences in $G_i$ between elements can be substantial (e.g., $G_C - G_N = 0.28\,eV$ atom$^{-1}$ at 300 K), at higher temperatures, differences in $G_i$ of >1 eV atom$^{-1}$ can result between solid and gaseous elements (e.g., $G_C - G_N = 1.12\,eV$ atom$^{-1}$ at 1200 K, Fig. 1b). In contrast to the elemental Gibbs energies, $G_i$, which are tabulated and thus require no computation or experiment to obtain, the Gibbs energies of solid compounds, $G^\delta$, are rarely tabulated and computationally demanding to calculate. Furthermore, assuming that all temperature-dependent effects can be captured by only including the elemental Gibbs energies and neglecting those of the solid compound results in an incomplete cancellation of errors and consequently inaccurate $\Delta G_f(T)$.

The temperature dependence of the thermodynamic properties of solids have often been assumed to be negligible relative to that of gaseous species[35]. That is, the Gibbs energy is generally assumed to be primarily entropic and principally due to vibrations such that the temperature dependence of the formation energies of solids is negligible. We examined this assumption for hundreds of solid compounds by comparing the difference between the experimental $\Delta G_f(T)$ and the approximate $\Delta G_f(T)$ that results from assuming negligible temperature dependence of the solid phase:

$$\Delta G_{f,app}(T) = \Delta H_f(298\,K) - \sum_{i=1}^{N} \alpha_i G_{i,gas}(T) \qquad (3)$$

Given a binary solid $AB$, if $A$ and $B$ are both solid at a given temperature, this assumption holds reasonably well and $\Delta H_f$ predicts $\Delta G_f(T)$ relatively accurately, e.g. with mean absolute errors of ~50 meV atom$^{-1}$ at 900 K (Fig. 1c). However, if either $A$ or $B$ are liquid at a given temperature, this error grows to ~100 meV atom$^{-1}$ at 900 K. Even more alarming is the error produced by this approximation if either $A$ or $B$ are gaseous at $T$, as is the case for oxides, nitrides, halides, etc. with mean absolute errors for $\Delta G_f(T)$ of ~200 meV atom$^{-1}$ at 900 K. In this approximation, the chemical potential, $G_i(T)$, of the gaseous element and the formation enthalpy, $\Delta H_f$, of the solid compound are taken from experiment and thus the larger error arises entirely from the missing quantity $G^\delta(T)$. The larger error that arises when an element is a gas or liquid, but not a solid, is due to the incomplete cancellation of the solid vibrational entropy of the elemental forms and the solid compound. That is, the distribution of phonon frequencies in the crystalline compound of $A$ and $B$ produce vibrational entropy $S_{AB}$ and if $A$ and $B$ are elemental solids, they too have solid vibrational entropies $S_A$ and $S_B$ where from Fig. 1c, we can presume in general: $S_{AB} \approx S_A + S_B$. However, when, for example, $A$ is a diatomic gas, the magnitude of the

frequencies of the molecular vibrations of $A$ are significantly larger and the incomplete cancellation of the vibrational entropy of $AB$ and $B$ leads to significant error as temperature increases.

**Descriptor identification and performance.** Because $\Delta H_f$ and $G_i(T)$ are readily obtained from tabulated calculated or experimental results, it is the lack of tabulated $G^\delta(T)$ which prevents the tabulation of $\Delta G_f(T)$ in computational materials databases (Eq. 2). The SISSO (sure independence screening and sparsifying operator) approach[28] was used to identify the following descriptor for $G^\delta(T)$:

$$G^\delta_{SISSO}(T)\left[\frac{eV}{atom}\right] = \left(-2.48*10^{-4}*\ln(V) - 8.94*10^{-5}\,mV^{-1}\right)T$$
$$+ 0.181*\ln(T) - 0.882$$
$$(4)$$

where $V$ is the calculated atomic volume (Å$^3$ atom$^{-1}$), $m$ is the reduced atomic mass (amu), and $T$ is the temperature (K). SISSO efficiently selects this descriptor from a space of ~$3 \times 10^{10}$ candidate three-dimensional descriptors, where the dimensionality is defined as the number of fit coefficients (excluding the intercept). A training set of 262 compounds with 2,991 ($T$, $G^\delta$) points was randomly selected from 309 inorganic crystalline solid compounds with experimentally measured $G^\delta(T)$ (Fig. 1a) and was used for descriptor identification. The remaining 47 compounds with 558 ($T$, $G^\delta$) points were reserved for testing. The descriptor performs comparably on the training and test sets with mean absolute deviations between the descriptor and experiment of <50 meV atom$^{-1}$ on both sets (Fig. 2). Notably, there is some $T$-dependence on the magnitude of residuals, with larger deviations as $T$ (and therefore the magnitude of $G^\delta$) increases (Supplementary Fig. 2). There are three plausible reasons for this: (1) the magnitude of $G^\delta$ being predicted increases so at fixed relative error, the magnitude of the residuals is larger, (2) the number of compounds with measured $G^\delta(T)$ decreases as $T$ increases, and (3) the physics dictating $G^\delta$ at high $T$ are more complex due to e.g., significant anharmonic vibrational effects that are less accurately captured by the simple model of Eq. (4). Approximately one-third of the compounds considered have measured $G^\delta$ (1800 K) and the mean absolute deviation (MAD) between $G^\delta_{SISSO}$ and $G^\delta_{exp}$ is found to increase from 53 to 92 meV atom$^{-1}$ from 1000 to 1800 K on the 47 compound test set. However, the relative MAD actually decreases from 14 to 11% over this same range on the test set, supporting reason (1) as a primary driver for the increasing residuals at elevated temperature. Violin plots of residuals for the training and test sets as a function of temperature are shown in Supplementary Fig. 2. More details of the approach used for descriptor identification can be found in Methods.

While a number of elemental and calculated properties were considered as inputs, it is notable that SISSO selects a descriptor dependent on only three quantities—temperature, atomic mass, and (calculated) atomic volume. The identification of these properties agrees well with intuition regarding the properties that most significantly affect the magnitude of vibrational entropy and free energy[36,37]. The phonon frequencies in a solid compound, $\omega$, are proportional to the force constant of the vibrational mode, $k$, and the reduced mass, $m$, of the vibrating atoms of the mode, with $\omega \sim \sqrt{k/m}$ in the harmonic oscillator approximation. As a mode's stiffness increases or its reduced mass decreases, its vibrational frequency increases, leading to a decrease in vibrational entropy and more positive Gibbs energies. This relationship is also apparent in the descriptor for $G^\delta(T)$, where $m$ is included directly and $V$ appears as a surrogate for $k$ (larger

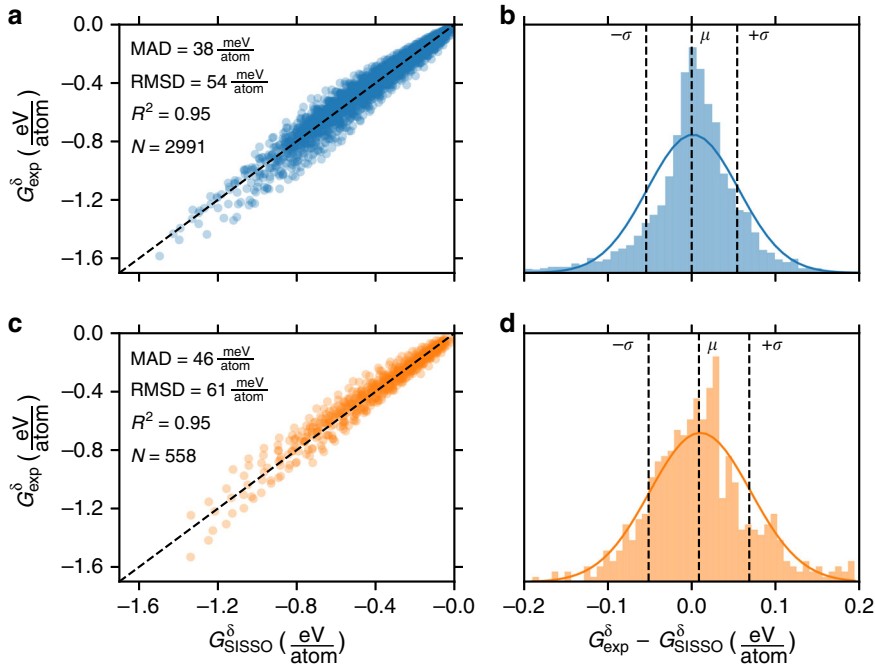

**Fig. 2** Descriptor performance. **a** Performance of the SISSO-learned descriptor (Eq. 4) on the training set. **b** Distribution of residuals between the SISSO-learned descriptor and experiment on the training set. **c** Performance of the SISSO-learned descriptor (Eq. 4) on the test set. **d** Distribution of residuals between the SISSO-learned descriptor and experiment on the test set. MAD is the mean absolute deviation, RMSD the root mean square deviation, $N$ the number of points shown, $\mu$ the mean deviation and $\sigma$ the standard deviation. The curved lines are normal distributions constructed from $\mu$ and $\sigma$

atomic volumes being associated with less stiff bonds or lower $k$). At constant $m$ and $V$, increasing temperature decreases $G^\delta$ when $-2.48 * 10^{-4} * \ln(V) - 8.94 * 10^{-5}\text{mV}^{-1} \leq 0.181\ln(T)/T$. This condition is uniformly satisfied for all 309 compounds in the training and test sets from 300 to 1800 K, reflecting the expectation of the negative temperature dependence of the Gibbs energy from fundamental thermodynamic expressions—e.g., $G = H - TS$. With $V$ and $T$ fixed, increases in $m$ result in more negative Gibbs energies, agreeing with the behavior of a harmonic oscillator for which $\omega$ depends inversely on mass and $G^\delta$ depends inversely on $\omega$. Finally, with $m$ and $T$ fixed, the descriptor (Eq. 4) indicates that $G^\delta$ becomes more negative for larger $V$ (for $V > 1$ Å³ atom⁻¹, i.e., all solid systems), in agreement with $V$ acting as a surrogate for the bond stiffness in the expression for the frequencies of a harmonic oscillator. Importantly, $V$ is the only structural parameter in Eq. (4) and therefore, at fixed composition (chemical formula), $G^\delta$ varies between structures (i.e., polymorphs) only as $V$ varies and $G^\delta(V)$ dictates that less dense structures of the same composition will have more negative $G^\delta$. Therefore, the prediction of polymorphic phase transitions is beyond the scope of this descriptor.

The quasiharmonic approximation (QHA) is commonly applied as an ab initio method for approximating $G$ (in practice, $G^\delta$)[13]. This approach typically requires a number of DFT calculations because the Helmholtz energy, including the electronic ground-state energy and the free harmonic vibrational energy, must be calculated as a function of volume (typically over a range of 10 or more volumes). Because of the high computational cost associated with QHA calculations, the number of structures with calculated $G$ is about 4 orders of magnitude less than the number of structures with calculated formation enthalpies, $\Delta H_f$. As an additional test set for the SISSO-learned descriptor for $G^\delta$, we compare our predictions to 131 compounds with tabulated $G^\delta$ in the PhononDB set which are not also in the experimental set compiled from FactSage used for training and testing the descriptor (Fig. 3a, b). For these

compounds, the descriptor agrees well with the ab initio values calculated using QHA, with a mean absolute deviation of 60 meV atom⁻¹. Notably, there is a nearly systematic underestimation of QHA-calculated $G^\delta$ by the descriptor with $G^\delta_{QHA} > G^\delta_{SISSO}$ for 98% of $(T, G^\delta)$ points in this set. Comparing QHA to experiment for an additional 37 compounds with experimentally measured $G^\delta$ available in FactSage reveals a similar systematic deviation with $G^\delta_{QHA} > G^\delta_{exp}$ for 94% of points (Fig. 3c, d). A number of factors likely contribute to the systematic offset between QHA and experiment including the approximations associated with the calculation (e.g., DFT functional and approximation to anharmonic vibrations), the neglect of additional contributions to the Gibbs energy including configurational and electronic entropy, and potential impurities or defects in the experimentally measured samples. It is notable that the deviation between $G^\delta_{QHA}$ and $G^\delta_{exp}$ is mostly systematic ($R^2 \sim 0.97$), so stability predictions based on convex hull phase diagrams constructed using ab initio $G^\delta_{QHA}$ should benefit from a fortuitous cancellation of errors, leading to even lower errors in practice than the already small deviation of 41 meV atom⁻¹ on average. Remarkably, for the same set of 37 compounds, our descriptor has lower mean absolute deviation from experiment than QHA (Fig. 3e, f) but does not exhibit this systematic underestimation of the magnitude of $G^\delta$ owing to its exclusive use of experimentally measured data for descriptor selection. While this magnitude of deviation for $G^\delta$ between experiment and prediction (using either QHA or the SISSO-learned descriptor) has been quoted as chemical accuracy (~1 kcal mol⁻¹) in the context of $\Delta H_f$[38], it is important to note that temperature-dependent predictions of stability using Gibbs formation energies, $\Delta G_f(T)$, will be affected by errors in both $G^\delta(T)$ and the temperature-independent $\Delta H_f$.

**Thermochemical reaction equilibria.** We combine our high-throughput model for the prediction of $G^\delta(T)$ with tabulated and readily available DFT-calculated $\Delta H_f$ and experimental Gibbs energies for the elements, $G_i(T)$ into Eq. (2) to enable the rapid

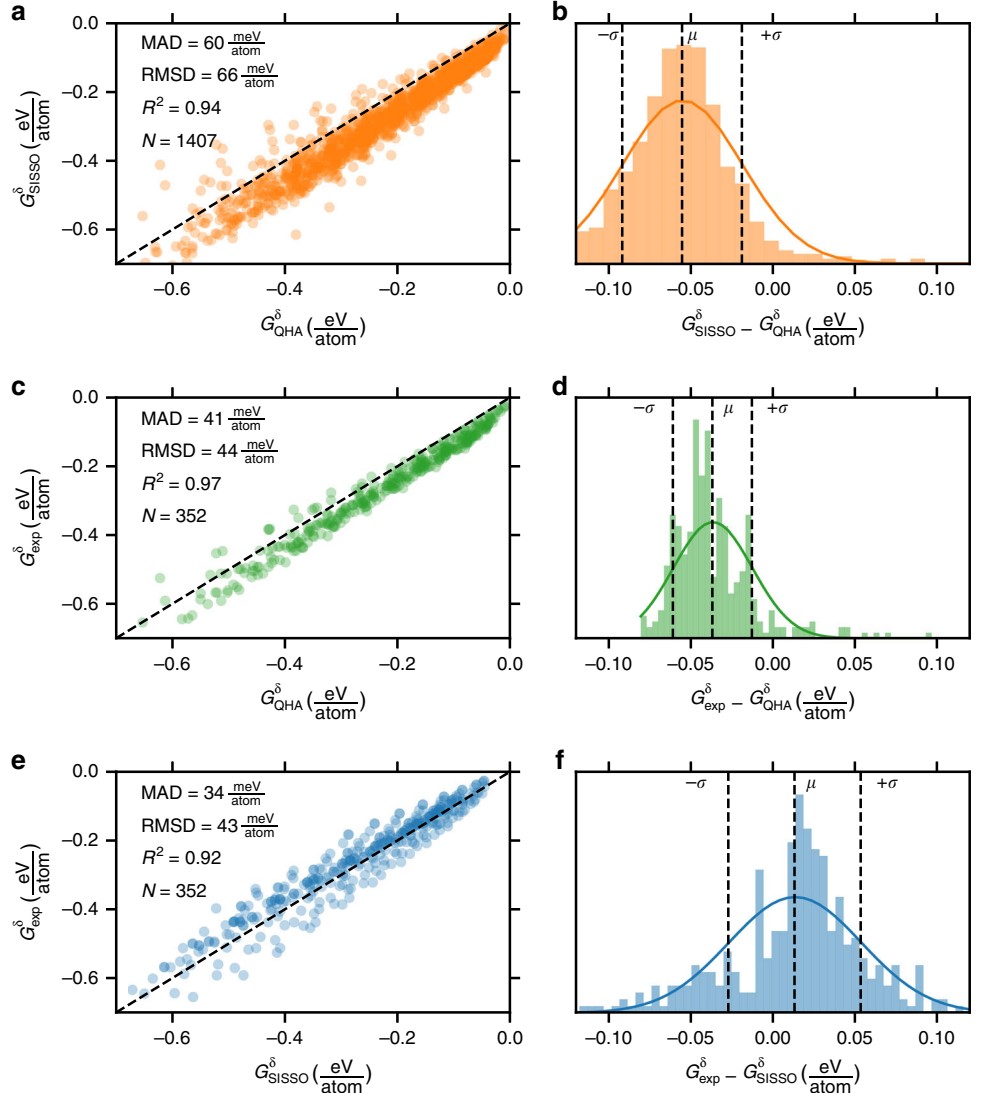

**Fig. 3** Benchmarking descriptor against ab initio methods. **a** Comparing the SISSO-learned descriptor to QHA for 131 compounds not in the experimental dataset used to train or test the descriptor. **b** Distribution of residuals shown in **a**. **c** Comparing QHA to experiment for 37 compounds which appear in both FactSage and PhononDB. **d** Distribution of residuals shown in **c**. **e** Comparing the SISSO-learned descriptor to experiment of these same 37 compounds shown in **c**. **f** Distribution of residuals shown in **e**. The annotation within each figure is provided in the Fig. 2 caption

prediction of $\Delta G_f(T)$ from a single DFT total energy calculation. Thus, reaction energetics, thermochemical equilibrium product distributions, and temperature-dependent compound stability can be assessed for the millions of structures currently compiled in materials databases. This unprecedented ability to rapidly predict reaction equilibria for reactions involving solid compounds is illustrated in Fig. 4 for a small set of example reactions. In Fig. 4a, the Gibbs energy of reaction, $\Delta G_{rxn}(T)$, which dictates the equilibrium spontaneity of any reaction event, is demonstrated for: the decomposition of SnSe[39], solar thermochemical hydrogen generation by the Zn/ZnO redox cycle[40], the carbothermal reduction of NiO to Ni[41], the oxidation of MoS$_2$[42], and the corrosion of CrN by water[43]. In each case, $\Delta G_{rxn}$ computed from the SISSO-learned descriptor for $G^\delta(T)$ agrees both qualitatively and quantitatively with $\Delta G_{rxn}$ resulting from the experimental values for $G^\delta(T)$. As a more sophisticated demonstration, Fig. 4b shows the equilibrium product distribution based on Gibbs energy minimization for the hydrolysis of Mo$_2$N to MoO$_2$ in the context of solar thermochemical ammonia synthesis[44]. In this analysis, Mo$_2$N and H$_2$O are placed in a theoretical chamber

at 1 atm fixed pressure and allowed to reach thermodynamic equilibrium with a set of allowed products—MoO$_2$, Mo, NH$_3$, H$_2$, and N$_2$—where the equilibrium product distribution at each temperature is that which minimizes the combined Gibbs formation energy of all species in the chamber. Even for this relatively complex system, the predicted product distribution based on the descriptor for $G^\delta(T)$ agrees both qualitatively and quantitatively with the product distribution calculated from the experimental $G^\delta(T)$. While this capability is demonstrated here to illustrate the utility of the identified descriptor for a few example reactive systems, this procedure is readily amenable for predicting reaction equilibria and product distributions in a high-throughput manner with numerous reacting species for a wide range of solid-state reactions. The accuracy of the descriptor-predicted reaction energies for new systems will be dependent not only on the effectiveness of $G^\delta_{SISSO}(T)$ to approximate $G^\delta_{exp}(T)$ but also on the extent to which DFT-predicted $\Delta H_f$ agrees with experiment as both parameters are required to obtain $\Delta G_f(T)$ (Eq. 2) and therefore $\Delta G_{rxn}(T)$.

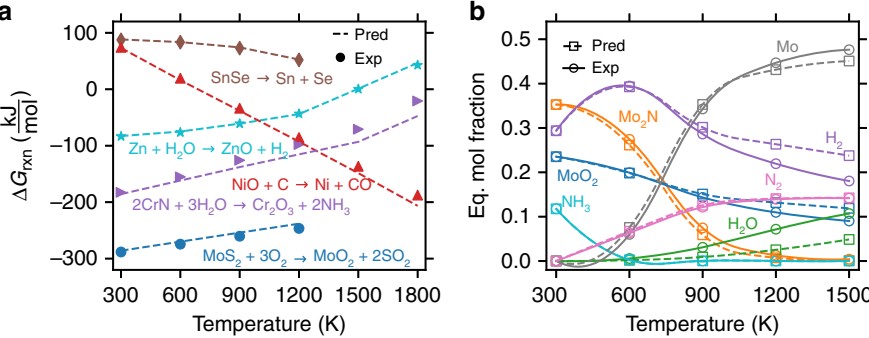

**Fig. 4** High-throughput reaction engineering. **a** A comparison of experimental reaction energetics (labels) to those predicted using the machine-learned descriptor for $G^\delta(T)$ (dashed curves). **b** Reaction product distribution between $MoO_2$, $Mo_2N$, $N_2$, $H_2$, $H_2O$, and $NH_3$ based on Gibbs energy minimization subject to molar conservation and fixed pressure of 1 atm. In both figures, pred applies the SISSO-learned descriptor to $G^\delta(T)$ of the solid phases and experimental data for all other components

**Effect of temperature and composition on material stability.** Beyond the investigation of solid-state reaction equilibria for a few example systems, we have also used the descriptor for $G^\delta(T)$ to compute phase diagrams to obtain broad insights into the temperature-dependent stability and metastability of thousands of known stoichiometric compounds. In particular, in the convex hull construction, formation energies, $\Delta G_f$, are plotted as a function of composition, and joined to produce the convex object of largest area. If $\Delta G_f$ of a composition lies above the convex hull, the composition is thermodynamically metastable and the vertical distance from the hull quantifies the magnitude of metastability of the compound, where larger distances indicate a greater thermodynamic driving force for decomposition of the metastable phase into stable phases. For the first time, temperature can be incorporated as a third axis in a high-throughput manner using $G^\delta_{SISSO}$ to produce $\Delta G_f(T)$ and assess the stability of compounds.

The Materials Project tabulates calculated structures for 29,525 compositions which also have reported ICSD numbers, suggesting that they have been realized experimentally[45]. Previous efforts to analyze temperature-independent metastability used $\Delta H_f$ as a surrogate for formation energy to predict that ~50% of all ICSD structures are metastable at 0 K[46]. We predict that ~34% of ICSD compositions are metastable in the absence of temperature effects —i.e., also using $\Delta H_f$. An important distinction between structures and compositions is that if a given composition has more than one known structure, all structures except the ground state at a given set of thermodynamic conditions are, by definition, metastable under those conditions. As such, in our analysis, we consider all structures of the 29,525 compositions, but only report statistics for the ground-state structures at each temperature (Figs. 5 and 6).

The fraction of compositions that are thermodynamically metastable remains nearly constant up to ~900 K where the competing effects of the elemental phases (Fig. 1b) lead to increasing compound destabilization with temperature (Fig. 5a). The fraction of compounds which move onto and off of the convex hull with temperature are also quantified relative to those that are predicted to be metastable and stable at 0 K. If a given composition exhibits no stable structures at 0 K (i.e., ~34% of the ICSD), it is unlikely that any of these structures become thermodynamic ground states at higher temperatures. In fact, only 1,602 of the 10,001 0 K metastable compositions are found to be stabilized when temperature is increased up to 1800 K. For the 1,602 compounds which are 0 K metastable but that come onto the hull to become stable at elevated temperature, the magnitude of their 0 K metastability is quantified in Fig. 5b. In general, compounds must lie very near to the hull at 0 K to have a chance

of thermal stabilization at $T > 0$ K. Even for compounds which become thermodynamic ground states at 1200 K, we find their metastabilities at 0 K to be typically <15 meV atom$^{-1}$ and thus thermal stabilization is often not the active mechanism in the high-temperature synthesis of solid compounds.

It is well known that metastable structures are often accessed experimentally, as indicated by the significant fraction of ICSD structures which are realized, but predicted to be metastable across this wide temperature range. A number of routes exist for accessing metastable structures, such as non-equilibrium synthesis conditions and alloying. In these cases, the magnitude of the metastability of these non-equilibrium structures indicates the driving force to convert to one or more stable phases, which is a critical consideration in materials processing and successful application of the material at operating conditions. Given the pool of metastable compositions in the ICSD, a Gaussian kernel density estimate is constructed based on the magnitude of metastability, $\Delta G_d$, and evaluated as a function of temperature (Fig. 5c) and composition (Fig. 6). At 0 K, 54% of metastable (but synthesized) compounds are >25 meV atom$^{-1}$ above the convex hull, 39% are >50 meV atom$^{-1}$, and 26% are >100 meV atom$^{-1}$ above the hull. These results provide some quantification for the false negative rate that is incurred by the ~25–100 meV atom$^{-1}$ heuristic error bars of materials screening approaches where compounds are typically allowed to survive stability screening if they are thermodynamically stable or within ~25–100 meV atom$^{-1}$ of metastability[46–49]. This range has been justifiably augmented in some cases, for example, in the search for novel 2D materials, which are by definition metastable, where the range has been expanded to e.g., 150 meV atom$^{-1}$[50]. Recent work has also shown that the 0 K energy of amorphous phases can provide an upper bound on the metastability of compounds that can be synthesized[48]. At low temperatures, the distribution of metastability is mostly constant with a median metastability of 43 meV atom$^{-1}$ at 900 K, suggesting that increasing the temperature from room temperature to 900 K results in only a small thermodynamic penalty of ~20 meV atom$^{-1}$. Above this temperature, many competing elemental phases undergo phase changes, leading to destabilization of compounds and a median metastability of 113 meV atom$^{-1}$ at 1800 K. This provides rationale for the viability of high-temperature solid-state synthesis approaches where increasing the temperature enables atomic rearrangements to overcome kinetic barriers while maintaining the desired structure as a thermodynamically accessible metastable state.

In addition to the temperature dependence of metastability, accessible compound metastability is also composition-dependent, as shown in Fig. 6. At 0 K, compounds comprised of most

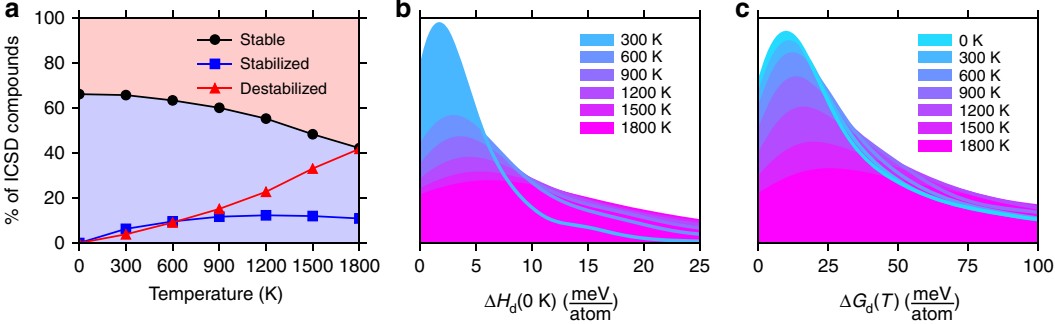

**Fig. 5** Survey of temperature-dependent (meta)stability. **a** Fraction of ICSD compositions which are thermodynamic ground states (black), fraction of 0 K metastable compositions which are stable at $T$ (blue), fraction of 0 K stable compositions which are metastable at $T$ (red), **b** Gaussian kernel density estimate of the 0 K decomposition enthalpy for ICSD compositions which are thermodynamically metastable at 0 K but stable at $T$, **c** Gaussian kernel density estimate of the Gibbs decomposition energy at $T$ for metastable compounds at each $T$. See Methods for additional details regarding this analysis

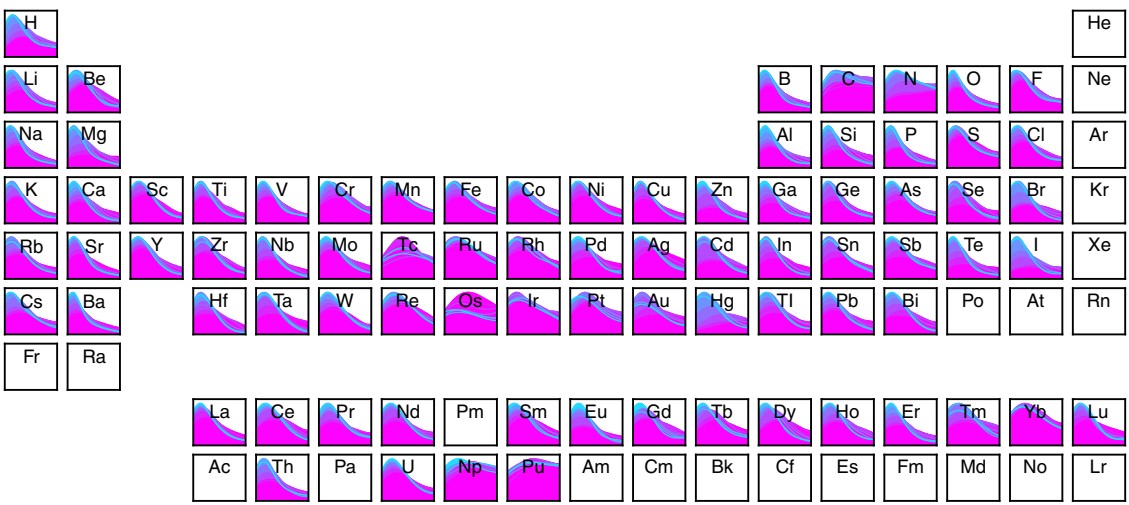

**Fig. 6** Composition-dependence of metastability. Elemental partitioning of the results shown in Fig. 5c by compounds containing element $X$. Each $x$-axis spans 0 to 100 meV atom$^{-1}$ and the colors align with the legend as shown in Fig. 5c

elements have a similar distribution of metastabilities to the overall distribution shown in Fig. 5c, with a few notable exceptions, particularly compounds containing carbon or nitrogen. For carbides and nitrides, the median metastabilities at 0 K are 144 meV atom$^{-1}$ and 109 meV atom$^{-1}$, more than five times the median metastability of all other compounds in the ICSD at 0 K (20 meV atom$^{-1}$). This prevalence of enhanced accessibility of metastable states was previously recognized for nitrides at 0 K and attributed to high cohesive energy which enables metastable configurations to persist[46,51]. The consequences of the high cohesive energies of these materials is low self-diffusion coefficients or high barriers to atomic rearrangement resulting from the tendency of the not-so-electronegative anions, carbon and nitrogen, to form mixed covalent/ionic bonds with electropositive and weakly electronegative elements across the periodic table.

Despite the similar metastability behavior of carbides and nitrides at low temperature, we find that temperature has a dramatically different effect on these two classes of compounds, with nitrides rapidly destabilizing by moving away from the hull and broadening their metastability distribution relative to carbides. The increases in median metastability for carbides and nitrides from 0 to 1800 K are 144 meV atom$^{-1}$ and 231 meV atom$^{-1}$, respectively. This can be attributed to the tendency for entropy to stabilize gaseous elemental nitrogen (i.e., $N_2$) with

temperature much more rapidly than solid elemental carbon (i.e., graphite). This creates the considerable high-temperature metastability difference that likely plays a critical role in enabling the synthesis of metastable carbides from amorphous precursors, where the lower thermodynamic driving force for phase separation of carbides at high temperature enables the persistence of higher energy amorphous precursor phases and increased thermal energy required to activate crystallization kinetics. The remarkable metastabilities exhibited by carbides and nitrides relative to other classes of materials provide chemical design principles for hindering atomic rearrangements and point towards these underexplored spaces for the discovery of highly metastable materials which are likely synthesizable.

## Discussion

Open materials databases are populated with millions of DFT-calculated total energies and formation enthalpies which have been used extensively for the design and discovery of new materials. However, critically lacking from these databases is the effect of temperature on the thermodynamics of these materials. To address this challenge, we have developed a simple and accurate descriptor for the Gibbs energy of inorganic crystalline solids, $G^\delta(T)$, using the SISSO approach. This low dimensional and physically interpretable descriptor reveals the main drivers for $G^\delta(T)$ to be the mass of the elements which comprise the

compound and the volume those atoms occupy in the material, agreeing well with the expectation from fundamental physical expressions and prior work quantifying the magnitude of vibrational entropy in solids. Remarkably, using only these parameters and temperature, the Gibbs energy can be predicted with accuracy comparable to the ab initio QHA approach up to at least 1800 K. Our descriptor for $G^\delta(T)$ can be readily applied to any of the more than one million structures with tabulated DFT total energy, enabling the high-throughput prediction of temperature-dependent thermodynamics across a wide range of compositions and temperatures.

Utilizing this descriptor, we demonstrate the accurate prediction of reaction energetics for a number of solid-state reactions, including a reaction network of several competing reactions in the context of thermochemical ammonia synthesis. This demonstrates how the descriptor can be incorporated with existing materials databases and tabulated thermochemical data for non-solids to predict the equilibrium products for an arbitrary reaction as a function of temperature. By applying the descriptor to ICSD compounds in the Materials Project database, we obtain the first comprehensive look at materials stability, providing a quantitative determination of how narrowly nature and inorganic synthesis have explored far-from-equilibrium materials and providing guidance for compositional considerations in realizing new metastable materials. While thermodynamic stability is the primary criterion used in high-throughput computational screening of materials to predict the likelihood of a given material being synthesizable, the interplay of thermodynamics with several other criteria, such as kinetics and non-equilibrium process conditions or starting precursors, exhibit a stronger influence over the synthesizability of materials, and currently, there is not a universal and well-defined metric for synthesizability[3,46,48,52-54]. Importantly, the ~50 meV atom$^{-1}$ resolution in predicting $G^\delta(T)$ achieved by our descriptor exceeds the accuracy of the computational methods that currently predict and populate $\Delta H_f$ in materials databases. Therefore, when combining $G^\delta(T)$ with $\Delta H_f$ to determine the Gibbs formation energy, $\Delta G_f(T)$, errors in these approaches will be additive, emphasizing the need for new or beyond-DFT methods to calculate $\Delta H_f$ when extremely high accuracy is required for a given application. However, there are many examples where DFT-computed $\Delta H_f$ was used successfully to realize new materials[55-57] and the incorporation of temperature effects using the SISSO-learned descriptor for $G^\delta(T)$ should only enhance these efforts.

## Methods

**Data retrieval.** Gibbs energies were extracted from the FactSage[29] experimentally determined thermochemical database for 309 solid compounds and from the PhononDB[16] ab initio calculated thermochemical database for 131 additional solid compounds (12 hydrides, 26 carbides, 31 nitrides, 104 oxides, 43 fluorides, 26 phosphides, 47 sulfides, 36 chlorides, 17 arsenides, 30 selenides, 40 bromides, 18 antimonides, 26 tellurides, 34 iodides; 313 binary compounds, 126 ternary compounds, and 1 quaternary compound—see Supplementary Data 1 for all compounds) and 83 elements. Compound data were extracted only at temperatures where the 298 K solid structure persists as reported in FactSage. Elemental data was obtained for the phase (solid crystal structure, liquid, or gas) with the minimum Gibbs energy at a given temperature. Because the 298 K enthalpy of formation, $\Delta H_f$ is well-predicted for compounds using high-throughput DFT along with appropriate corrections[30-33] and readily available for millions of structures in existing materials databases, the Gibbs energy was referenced with respect to $\Delta H_f$ (Eq. (1)).

**Feature retrieval.** Nine primary features were considered for this work—five tabulated elemental properties (electron affinity, first ionization energy, covalent radius, Pauling electronegativity, and atomic mass) extracted from pymatgen[58] and WebElements (http://www.webelements.com); two calculated properties (atomic volume and band gap) extracted from the Materials Project database; one experimental property ($\Delta H_f$), and temperature. The five tabulated elemental properties were formulated into compound-specific properties using each of three transformations. For elemental feature, $x$, we define three forms of averaging—the

stoichiometrically weighted mean (avg), the stoichiometrically weighted harmonic mean, akin to the reduced mass (red), and the stoichiometrically weighted mean difference (diff):

$$x_{\mathrm{avg}} = \frac{1}{\sum_{i=1}^{N} \alpha_i} \sum_i^N \alpha_i x_i \tag{5}$$

$$x_{\mathrm{red}} = \frac{1}{(N-1)\sum_{i=1}^{N} \alpha_i} \sum_{i \neq j}^N \left(\alpha_i + \alpha_j\right) \frac{x_i x_j}{x_i + x_j} \tag{6}$$

$$x_{\mathrm{diff}} = \frac{1}{(N-1)\sum_{i=1}^{N} \alpha_i} \sum_{i \neq j}^N \left(\alpha_i + \alpha_j\right) |x_i - x_j| \tag{7}$$

where when considering a compound, $A_a B_b C_c$, we define $\boldsymbol{\alpha}$ as the vector of coefficients $[a, b, c]$ and $N$ as the length of $\boldsymbol{\alpha}$. For example, for CaTiO$_3$, $\boldsymbol{\alpha} = [1,1,3]$ and $N = 3$.

**Descriptor identification.** The SISSO approach[28] was applied to identify the descriptor for $G^\delta$ shown in Eq. (3) using 262 of the 309 compounds from FactSage with experimentally measured $G^\delta$. To identify this descriptor an initial feature-space, $\Phi_0$, included 19 features—the five tabulated elemental properties mapped onto each of the three functional forms (Eqs. 5–7), along with the linear forms of atomic volume, band gap, formation enthalpy, and temperature. Two iterations of descriptor construction were performed using an operator space of $[+, -, |-|, *, /,$ exp, ln, $^{-1}, ^2, ^3, ^{0.5}]$. Candidate descriptors were constructed by iteratively applying these operators to $\Phi_0$ while conserving the units of constructed features. The first iteration of descriptor construction yielded a space, $\Phi_1$, with ~600 candidate descriptors and the second iteration a space, $\Phi_2$, of ~600,000 candidate descriptors. SISSO was then performed on $\Phi_2$ with a subspace size of 2,000 and three descriptor identification iterations, thereby producing the three-dimensional (3D) descriptor (i.e., three fit coefficients not including the intercept) in Eq. (4). In the first iteration, sure independence screening (SIS) was used to select the 2,000 descriptors $S_{1D}$ from $\Phi_2$ having the highest correlation with $G^\delta$. Within $S_{1D}$, $\ell_0$-norm regularized minimization, SO($\ell_0$), was used to identify the best 1D descriptor. This 1D descriptor is then used to predict the training set and the array of residuals, $R_1$, is generated from this prediction. Now with $R_1$ as the target property (instead of $G^\delta$), SIS identifies a new subspace $S_{2D}$ of 2,000 additional descriptors. SO($\ell_0$) then selects the best-performing 2D descriptor from $S_{1D} \cup S_{2D}$ and $R_2$ is generated as the residuals using this 2D descriptor to predict the training set. This procedure is repeated a third time to yield the 3D descriptor shown in Eq. (4). Therefore, this descriptor is selected among a space of $\binom{6000}{3}$ or ~$3 \times 10^{10}$ candidate 3D descriptors.

Importantly, all aspects of the SISSO selection algorithm were performed on the training set of 262 compounds with experimentally measured Gibbs energies, leaving an excluded test set of 47 compounds with experimentally measured Gibbs energies in reserve to evaluate the predictive quality of the selected descriptor (Fig. 2). An additional 131 compounds with QHA-calculated $G^\delta(T)$ not present in the training or test sets were also compared with the SISSO-learned $G^\delta(T)$ (Fig. 3).

**Descriptor sensitivity.** While the random splitting of the experimental set into training and test sets was performed only once, comparing the relevant properties for each set reveals that they are statistically similar, suggesting the model and SISSO process would yield similar results for an arbitrary random split of the experimental set (Supplementary Fig. 3). To assess the robustness of the model on diverse training and test sets, we repeated the random split of the experimental set 1,000 times and evaluate the performance of Eq. (4) on each set. The MAD spans 37–42 meV atom$^{-1}$ on the 85% training set and 26–54 meV atom$^{-1}$ on the 15% test set, demonstrating that the reported 38 meV atom$^{-1}$ for training and 46 meV atom$^{-1}$ for testing (Fig. 2) are not outliers. As an added demonstration, the random split of the experimental set and subsequent SISSO selection process was repeated 12 times. In 10/12 runs, the descriptor shown in Eq. (4) appears in the top 3,000 of ~$3 \times 10^{10}$ models evaluated (top ~0.00001%) in terms of root mean square deviation (RMSD) on the training set. Notably, there are many cases where very slight deviations of Eq. (4) also appear in the top models—e.g., replacing ln(T) with $T$ or $T^{0.5}$. To validate the significance of the three features that comprise the descriptor—temperature, reduced mass, and atomic volume—we assess what fraction of the top 3,000 models contain these features for each of the 12 random train/test splits. Temperature is found to occur in 100% of the top models for each of the 12 random splits. Reduced mass and atomic volume each appear in ~86% of the top 3,000 models on average over the 12 random splits. This analysis was conducted on only the very best models (top ~0.00001%) and reveals the significance of these three properties in predicting $G^\delta$ to be robust to the random split of the experimental data used to train and test the descriptor. Notably, the first term in Eq. (4), $T\ln(V)$, appears as the feature with the highest correlation with $G^\delta$ in all of the 12 random train/test splits.

**Comparing to QHA**. QHA-calculated $G(T)$ was extracted from the 2015 version of PhononDB[16] for all compounds with calculated thermal properties. Because a number of approximations are used to calculate $\Delta H_f$ from DFT calculations, to isolate the temperature-dependent Gibbs energy for comparison to our descriptor, $G^\delta_{QHA}(T)$ was calculated as $G^\delta(T) = G(T) - G(0\,K)$.

**Stability analysis**. For the generation of Figs. 5 and 6, all 34,556 entries (structures) in the Materials Project which have reported formation energies and ICSD numbers were retrieved. For each entry, the temperature-dependent formation energy was calculated as follows:

$$\Delta G_{f,pred}(T) = \begin{cases} \Delta H_{f,MP}, \, T = 0\,K \\ \Delta H_{f,MP} + G^\delta_{SISSO}(T) - \sum_i^N \alpha_i G_{i,exp}(T), \, T \neq 0\,K \end{cases} \quad (8)$$

FactSage elemental energies were used as $G_{i,exp}$. For all entries, $\Delta G_{f,pred}(T)$ was evaluated at 0, 300, 600, 900, 1200, 1500 and 1800 K. To avoid overweighting the analysis to compounds which have many polymorphs, the lowest (most negative) $\Delta G_{f,pred}(T)$ was retained for the analysis at each temperature and for each unique composition (chemical formula). This resulted in 29,525 unique compositions from 34,556 structures with ICSD numbers and reported formation energies in Materials Project. To avoid potentially spurious entries in the ICSD, only the lowest 90% of metastable compositions (with respect to the Gibbs decomposition energy, $\Delta G_d$) were considered. Python was used to construct all possible convex hull phase diagrams and quantify $\Delta G_d$.

**Structure considerations**. For training, we used 0 K ground-state structures (and magnetic configurations) reported in Materials Project. From this calculation result, we retrieved the volume (per atom) that is then used at all temperatures to generate $G^\delta(T)$ as shown in Eq. (4). For a given composition, one could compute $G^\delta(T)$ for any number of structural or magnetic configurations and compare the $G(T)$ that results. For the purposes of training and testing, we consider only the calculated ground-state because this is likely the approach that would be used in practice for the application of the model to new materials which have available calculated but not experimental data.

**Application of the descriptor**. To obtain the Gibbs formation energy for a given structure, one must first perform a DFT total energy minimization of the structure. From this, the atomic volume is determined as the volume of the calculated cell divided by the number of atoms in the calculated cell. $G^\delta$ can then be computed by Eq. (4). Calculating the Gibbs energy, $G(T)$, using Eq. (1) requires the formation enthalpy, $\Delta H_f$, calculated using DFT. If the analysis of interest concerns only one composition (chemical formula), then this is the final step and the relative energies of all structures with this composition can be compared using $G(T)$. If the analysis of interest considers various compositions (e.g., for convex hull stability or thermochemical reaction analysis), the elemental energies must be subtracted to obtain the Gibbs formation energy, $\Delta G_f(T)$ by Eq. (2). Notably, $\Delta H_f$ and volumes calculated by DFT are tabulated for many thousands of structures and the elemental $G(T)$ are also tabulated for at least 83 elements. An important point is that users of the descriptor for $G^\delta(T)$ are free to generate $\Delta H_f$ and volumes for any number of structural or magnetic configurations for a given composition and compare how $G(T)$ might be sensitive to the changes in structure and magnetism.

**Extension to new materials**. On the experimental training set of 262 compounds, the mean absolute deviation between experiment and the descriptor is 38 meV atom$^{-1}$ (Fig. 2). This increases slightly to 46 meV atom$^{-1}$ (Fig. 2) on the experimental test set and to 60 meV atom$^{-1}$ on the computed (QHA) test set (Fig. 3). The residuals with respect to experiment are also mostly normally distributed, suggesting no systematic error in the model. The performance on the test set compounds is a demonstration of validated prediction accuracy or uncertainty on new predictions. These approximate error bars can be expected on additional new predictions to the extent that the sets used for training and testing are comparable to the new materials being predicted. The set we use for training and testing is quite diverse—83 unique elements, binaries and multinaries, magnetic and nonmagnetic, metallic and insulating, etc. Additionally, the descriptor is relatively simple, having only four fit parameters (including the intercept) and three features (properties) that it depends upon. However, it has not been benchmarked for non-stoichiometric compounds or compounds with defects. For example, one could not expect to obtain the temperature-dependent defect formation energy using our descriptor because this was not benchmarked. Our model is also not capable of predicting the melting point of compounds. $G^\delta(T)$ is for the solid phase and can be obtained even well above a compound's melting point, where the liquid phase has more negative Gibbs energy. As alluded to in the main text, the extension of the descriptor to correctly predict polymorphic phase transitions or temperature-driven magnetic transitions is not practical because the descriptor depends only on the mass, density, and temperature and the magnitude of the energy change for these transitions is typically smaller than the expected error bars of the descriptor.

We report substantial evidence that the descriptor is predictive for stability of compounds relative to one another and for the prediction of thermochemical reaction equilibria over a wide range of stoichiometric solid compounds with a diverse set of chemical and physical properties.

## Data availability

Data (via public repository), code, and associated protocols are available in a github repository (github.com/CJBartel/predict-gibbs-energies) corresponding to the implementation and application of the model as described within this work.

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

## Acknowledgements

We acknowledge Runhai Ouyang and Luca Ghiringhelli for useful discussions regarding SISSO. This work was supported by the U.S. Department of Energy, Office of Science, Office of Basic Energy Sciences, as part of the Energy Frontier Research Center "Center for Next Generation of Materials by Design" under contract no. DE-AC36-08GO28308 to the National Renewable Energy Laboratory (NREL). C.J.B gratefully acknowledges support from the NSF Graduate Research Fellowship Program under Grant DGE 114803 and from the Department of Education GAANN program. C.B.M., A.M.H., and S.L.M. gratefully acknowledge support from award no. CBET-1806079 and award no. CBET-1433521, which was co-sponsored by the National Science Foundation, Division of Chemical, Bioengineering, Environmental, and Transport Systems, and the DOE, EERE, Fuel Cell Technologies Office and from DOE award EERE DE-EE0008088, sponsored by the DOE, EERE, Fuel Cell Technologies Office.

## Author contributions

All authors provided feedback and input to the research, analysis and manuscript. The project was conceived by C.J.B., A.M.D., V.S. and A.M.H., and directed by C.B.M. and A.M.H. C.J.B. performed the materials informatics analysis under the guidance of C.B.M. and A.M.H., with assistance from J.R.R., S.L.M., and A.W.W. in data curation and W.T., V.S. and S.L. in developing the discussion of materials stability and synthesis. All authors reviewed and commented on the manuscript.

## Additional information

**Competing interests:** The authors declare no competing interests.

