## [Peer Review File · Nature Communications]

Reviewers' comments:

Reviewer #1 (Remarks to the Author):

In this paper, Bartel et al use a data-driven method to predict and tabulate the Gibbs energy for ~30,000 stoichiometric inorganic compounds curated from the ICSD. They employ a recently developed sure independence screening and sparsifying operator (SISSO) approach that builds a linear regression model (in its functional form) between the experimental Gibbs free energy data (response) of 309 compounds and descriptors from high-throughput DFT calculations. The independent variables of the linear model can take non-linear transformations and the authors have a brute-force way to explore the non-linear transformation and reduce the dimensionality. The SISSO approach identifies Temperature, Volume (from DFT) and atomic mass as important descriptors for predicting the Gibbs energy from screening more than a million possibilities, which is impressive. The authors also validate their regression model using an independent validation set, where the dataset is taken from a computational Phonon database that contains 109 compounds built from DFT Quasi-Harmonic Approximation phonon calculations. They report an error of 50 and 60 meV/atom for the training (experimental) and validation (Phonon, computational) datasets, respectively. They then apply their model to explore the Thermochemical reaction equilibria of a few systems and finally, apply it to predict the Gibbs energy for 30,000 inorganic compounds from ICSD. They also discuss some of the statistics and key implications of their work.

This is an interesting work and there is a need for developing an approach that can reliably predict Gibbs free energy of solids. The approach described in this paper also goes beyond the prediction capabilities of the current high-throughput DFT databases and the fact that it uses DFT data for feature construction also indicate a symbiosis between the community-developed database efforts and the developed mathematical model. However, I have several questions about this work that I hope will improve the manuscript and I enumerate them below:

- The interplay between thermodynamics and kinetics determines the synthesizability of a compound. Without providing any data on reaction pathways and the activation barrier for each of the pathway, I am not convinced that the Gibbs energy can be directly related to synthesizability. The authors should reconsider the usage of the term synthesizability in this work. They use the term only sporadically and it is giving me a perception that the authors are trying to oversell the work. If the authors are insistent on the language (synthesizability), then they should provide a solid justification. On the other hand, the argument about thermodynamic stability is justified.
- It was difficult for me to follow some of the notations in the paper. For example, on Page 3 the authors introduce H^Δ , but it is not found in any of the equations. On Page 2, it is not clear if $\Delta G_f(T)$ and $\Delta G(T)$ are the same. On Page 4, equation 4 has $\Delta G_{\{f,app\}}(T)$. What is meaning of the app subscript? I do not want to assume its meaning and create my own interpretation. Therefore, it is confusing and very difficult to follow the paper. I would request the authors to carefully proofread the article and check for the notations.
- For training their SISSO model, the authors used 262 compounds with 2,991 (T,G $^\Delta$) chosen randomly. But the test set of 47 compounds contain only 558 (T,G $^\Delta$). What is the difference between 2,991 and 558? It is not discussed in the paper.
- It is unclear if the Volume (from high-throughput DFT) chosen by the authors also correspond to the same structure for which the experimental free energy is measured. They should add space group and the magnetic state to their Table S1 (at least for the experimental data from FactSage).
- Since the experimental free energy data span a wide temperature range, did the authors also take into account the changes in the crystal structure brought about by the structural or magnetic phase transitions in the considered temperature range? For example, take CaTiO₃. The authors use CaTiO₃ (from FactSage) in their training set and they give a value of 1500 for the T_{max}. It is known that CaTiO₃ undergoes structural phase transition as a function of temperature (Ali et al. Journal of Solid State Chemistry vol. 178, pages 2867-2872, year 2005). How did the authors

account for phase transformations (in general) in their model? This is not discussed in the paper.

-- A related question then is the following: In their 309 compounds from FactSage, how many have one or multiple structural or magnetic phase transitions within their reported T_{\max} value? Do they have any statistic about this point?

-- Is it problematic or a significant limitation of their approach if the training data does not describe or capture phase transformations? I can immediately see that it will be difficult to choose a volume if a compound undergoes phase transformation as a function of temperature. This should also affect the number of atoms in the unit cell and the phonons dispersion curves (QHA phonons).

-- It appears that the authors have taken the Volume data from the Materials Project database. From Table S1, I also find several transition metal oxides. Can the authors comment on the importance of magnetism in the choice of volume as a descriptor? This is an important point because the experimental Gibbs free energy for an antiferromagnetic compound should have the volume information for the same crystal structure and magnetic configurations from DFT. Otherwise, the mathematical model makes no physical sense. Can the authors clarify?

-- In the SISSO method, the authors split the experimental training data into 262 and 47 for training and testing, respectively. The SISSO then identifies a functional form for predicting the Gibbs energy that contains volume, temperature and mass as descriptors. The authors then go on to justify the meaning behind this relationship, which is very interesting. My question is the following – if the authors randomly choose different sets of samples with the same 262 vs 47 split from using the same 309 experimental data (at least they should explore five sets of random sampling), would they get the same mathematical relationship every time (including the functional form)? I am curious to understand the robustness or sensitivity of their mathematical model to the training data. If they get different mathematical models with a different set of independent variables, what are the implications? How will the errors on test set and validation set (DFT-QHA Phonon database) behave or change every time for each of those random samples? In the absence of these assessments, I do not think their SISSO model building is complete.

-- The section on the “effects of temperature and composition on stability”, where they are applying their training SISSO model to predict the Gibbs energy for ~30,000 compounds, is also problematic. The overlap between the 309 compounds for which the experimental Gibbs energy is known and the ~30,000 compounds is not clear. In other words, are crystal structure, chemical elements, Valence states of transition metal ions and their magnetic configurations etc, in both the 309 and 30,000 data samples identically distributed? It appears that the authors are extrapolating their results through learning from a small dataset of 309 compounds and applying it to a much larger set of 30,000 compounds. In the absence of experimental validation or uncertainty quantification, how can I trust their predictions?

-- Under what circumstance does their SISSO model fail? Where can the predictions from the SISSO model be trusted? And more importantly, where does these models fail? If I cannot trust the model, what is the point of tabulating the Gibbs energy data for one million structures? There is no discussion about these crucial points.

-- The section on “Thermochemical reaction equilibria” is interesting and may have important implications. From Figure 4, it appears that their models are very accurate. A brief discussion on the limitations can help the readers.

-- I would also like to bring to the attention of the authors a couple of recent works on machine learning and materials discovery, where the Gibbs free energy data was not used for prediction of new compounds: Ren et al Sci Adv vol. 4, eaaq1556, year 2018 and Balachandran et al Nat Commun vol. 9, article number: 1668, year 2018.

Given these major concerns, I do not recommend this manuscript for publication in the Nature Communications journal.

Reviewer #2 (Remarks to the Author):

The paper describes a new descriptor for estimation of the difference between the temperature dependent Gibbs free energy and the standard Helmholtz enthalpy. The descriptor is designed by the aid of machine learning techniques and represents a simple function of atomic volume, the reduced atomic mass and temperature. In general, it would be beneficial to have a simple and accurate way to estimate the thermodynamic parameters of solid compounds from basic properties such as volumes, masses and constituent elements. It is not a new idea and in the last at least 100 years there were several related studies published (e.g. Latimer, J. Am. Chem. Soc. 43, 818 (1921)). Reading carefully the submitted manuscript I am not convinced that this study provides a "magic" formula that substantially improves the accuracy of these simple predictions and that could be used for precise thermodynamic considerations of solid compound stabilities. I thus do not see significant advancement in the prediction of thermodynamic parameters of solid compounds that would warrant rapid publication in Nature Communications. Nevertheless, the studies are definitely of interest to the thermodynamics community, but more appropriate for standard journals such as *Thermochimica Acta*. Below I give more details, justification and comments.

Detailed comments:

1) The title is a bit misleading as it suggests that a descriptor for prediction of the Gibbs free energy is given, while authors provide a formula for estimation of $G-H_f(298K)$ only.

2) Eq. 3: my understanding is that the terms besides DH_f should vanish at 298K, which is not true. These vanish at $T=324K$. The authors should check the correctness of the formula.

3) Uncertainty of Eq. 5: According to Fig. 2 the uncertainty in Eq. 5 (descriptor) is as large as 0.1 eV/atom ~ 10 kJ/atom. According to studies of Latimer, J. Am. Chem. Soc. 43, 818 (1921) or Spencer *Thermochimica Acta* 314, 1 (1998) the entropy contribution to the Gibbs free energy per atom is on average ~10 kJ/atom at 300K and 33 kJ/atom at 1000K. The uncertainty of Eq. 5 is thus very large, which in my opinion prevents quantitative estimates as claimed by the Authors. I have checked this on a few compounds reported in the Table submitted with the manuscript and the difference was indeed large.

4) Following comment 3, the Authors should provide a set of a few figures comparing predicted $G-H_f$ and the measured ones. Such experimental data exist for some compounds they consider.

Reviewer #3 (Remarks to the Author):

This paper presents an application of a previously published ML method for symbolic regression (SISSO) to the prediction of the temperature dependent Gibbs energy of crystalline solids. The paper is sound and technically interesting and should be published. I have some suggestions for small improvements below.

There is no obvious innovation in the machine learning as this appears to be an essentially unchanged application of the previously published SISSO method. However, I do not know of other work applying ML to the temperature dependent Gibbs energy. Further, SISSO and other symbolic regression techniques are not commonly applied to materials science and expanding the scope of what ML for materials science means is in the broader interests of the community. In this way, I expect this paper to be of interest to scientists already interested in ML for materials science.

I have two technical points I would like to see addressed.

Point 1

The results, such as those in Fig 4b, suggest a clear dependence of accuracy on temperature. A discussion of whether this is due to the model fitting or to extrapolation could be insightful. In particular, showing error plots such as in Fig 2 as a function of temperature would help answer the

question: "Does the model fit the high T data less well or is there something else going on?" but I'd like to see a discussion of this question in general.

Point 2

This paper does a poor job situating itself in the work of symbolic regression in general (though notably the original SISSO does not do a good job here either).

The authors compare to Eureka, but do not cite the most famous use of this Schmidt and Lipson

Distilling Free-Form Natural Laws from Experimental Data
<http://science.sciencemag.org/content/324/5923/81>

There are several past papers on symbolic regression in materials related problems (generally using genetic programming approaches, which was the traditional approach).

Makarov and Metiu

Fitting potential-energy surfaces: A search in the function space by directed genetic programming
<https://aip.scitation.org/doi/abs/10.1063/1.475421?journalCode=jcp>

Makarov and Metiu

Using Genetic Programming To Solve the Schrödinger Equation
<https://pubs.acs.org/doi/abs/10.1021/jp000695q>

Johnson

Evolutionary Algorithms Applied to Electronic-Structure Informatics: Accelerated Materials Design Using Data Discovery vs. Data Searching
https://lib.dr.iastate.edu/cgi/viewcontent.cgi?referer=https://www.google.com/&httpsredir=1&article=1291&context=ameslab_pubs

Sastry et al.

Genetic Programming for Multi-Timescale Modeling
<https://arxiv.org/abs/cond-mat/0405415>

And a variety of other publications such as these on methods applied to other physical-ish systems

Brunton et al.

Discovering governing equations from data by sparse identification of nonlinear dynamical systems
<http://www.pnas.org/content/113/15/3932/tab-article-info>

Koza, Keane, Rice

Performance improvement of machine learning via automatic discovery of facilitating functions as applied to a problem of symbolic system identification
<https://ieeexplore.ieee.org/document/298555/?reload=true>

There are also a few papers with very different ML methods (though these methods have not (to my knowledge) been applied to materials problems).

Kusner et al.

Grammar Variational Autoencoder
<https://arxiv.org/abs/1703.01925>

Dai et al.

Syntax-Directed Variational Autoencoder for Structured Data
<https://arxiv.org/abs/1802.08786>

Minor points on the presentation:

* Fig 1: I assume this is the data from FactSage and PhononDB that is used throughout the paper, but that is not obvious from the caption or the text.

* Fig 1b: I find this figure nearly impossible to read. The desaturated colors are hard to tell apart and the most desaturated ones are hard to see at all. I'm sure you can find a better way to display this data.

* Fig 1c: The error bars are not defined in the caption.

Reviewer:

Patrick Riley

Google Accelerated Science

pfr@google.com

Note for all reviewers: We very much appreciate your helpful comments and suggestions, especially regarding points of confusion in the manuscript. Reviewer comments are shown in *italic black text*, with our responses shown in blue. In the revised manuscript, changes are highlighted in yellow and also included in the response to the comment that motivated the change. Please also note that the removal of **Equation 2** from the main text has shifted the numbers of each subsequent equation.

Reviewer #1 (Remarks to the Author):

In this paper, Bartel et al use a data-driven method to predict and tabulate the Gibbs energy for ~30,000 stoichiometric inorganic compounds curated from the ICSD. They employ a recently developed sure independence screening and sparsifying operator (SISSO) approach that builds a linear regression model (in its functional form) between the experimental Gibbs free energy data (response) of 309 compounds and descriptors from high-throughput DFT calculations. The independent variables of the linear model can take non-linear transformations and the authors have a brute-force way to explore the non-linear transformation and reduce the dimensionality. The SISSO approach identifies Temperature, Volume (from DFT) and atomic mass as important descriptors for predicting the Gibbs energy from screening more than a million possibilities, which is impressive. The authors also validate their regression model using an independent validation set, where the dataset is taken from a computational Phonon database that contains 109 compounds built from DFT Quasi-Harmonic Approximation phonon calculations. They report an error of 50 and 60 meV/atom for the training (experimental) and validation (Phonon, computational) datasets, respectively. They then apply their model to explore the Thermochemical reaction equilibria of a few systems and finally, apply it to predict the Gibbs energy for 30,000 inorganic compounds from ICSD. They also discuss some of the statistics and key implications of their work.

This is an interesting work and there is a need for developing an approach that can reliably predict Gibbs free energy of solids. The approach described in this paper also goes beyond the prediction capabilities of the current high-throughput DFT databases and the fact that it uses DFT data for feature construction also indicate a symbiosis between the community-developed database efforts and the developed mathematical model. However, I have several questions about this work that I hope will improve the manuscript and I enumerate them below:

We are grateful to the reviewer for their positive comments and for posing several important questions. We provide detailed responses to each below.

A brief clarification on this initial comment – of the 309 compounds with experimental data we randomly selected 262 for training (mean absolute deviation (MAD) of 38 meV/atom) and 47 for testing (not included in training in any capacity) with MAD = 46 meV/atom (**Fig. 2**). 131 compounds with $G(T)$ computed using the quasiharmonic approximation were used for testing with a MAD of 60 meV/atom (**Fig. 3**). This is now clarified in the introduction.

Page 3: The descriptor is identified using experimental data²⁹ for 262 solid compounds and tested on a randomly chosen excluded set of 47 compounds with measured $G(T)$ and 131 compounds with first-principles computed¹⁶ $G(T)$.

-- *The interplay between thermodynamics and kinetics determines the synthesizability of a compound. Without providing any data on reaction pathways and the activation barrier for each of the pathway, I am not convinced that the Gibbs energy can be directly related to synthesizability. The authors should reconsider the usage of the term synthesizability in this work. They use the term only sporadically and it is giving me a perception that the authors are trying to oversell the work. If the authors are insistent on the language (synthesizability), then they should provide a solid justification. On the other hand, the argument about thermodynamic stability is justified.*

This is a fair point, and we agree with the reviewer. We have revised the concluding paragraph of the manuscript to highlight when stability (i.e., G) does not reflect synthesizability.

Page 14: While thermodynamic stability is the primary criterion used in high-throughput computational screening of materials to predict the likelihood of a given material being synthesizable, the interplay of thermodynamics with several other criteria, such as kinetics and non-equilibrium process conditions or starting precursors, exhibit a stronger influence over the synthesizability of materials, and currently, there is not a universal and well-defined metric for synthesizability.^{3,48,52-55}

-- It was difficult for me to follow some of the notations in the paper. For example, on Page 3 the authors introduce H^Δ , but it is not found in any of the equations. On Page 2, it is not clear if $\Delta G_f(T)$ and $\Delta G(T)$ are the same. On Page 4, equation 4 has $\Delta G_{f,app}(T)$. What is meaning of the *app* subscript? I do not want to assume its meaning and create my own interpretation. Therefore, it is confusing and very difficult to follow the paper. I would request the authors to carefully proofread the article and check for the notations.

Thank you for identifying these points of confusion. We now consistently include definitions of the notation at their first instance in the text to clarify what each symbol represents. We now only include the definition of H^δ in the caption of **Fig. 1a**.

Page 4 (caption): H^δ is the temperature-dependence of the enthalpy normalized to be zero at 298 K (**Equation S1**)

The best fit equation has been moved to the **Supplementary Information** as it is not critical to the results but may be of interest to some readers.

Page S2: $H^\delta(T) \left[\frac{\text{eV}}{\text{atom}} \right] = H(T) - H(298 \text{ K}) \approx (2.82 * 10^{-4})T[\text{K}] - 9.13 * 10^{-2}$ [S1]

$\Delta G(T)$ with no subscript does not appear in the manuscript. In our notation, Δ indicates change over some reaction. If that is a formation reaction, then we also include a subscript, *f*.

Regarding $\Delta G_{f,app}(T)$, the subscript, *app*, is an abbreviation for “approximation” and was defined in **Equation 4** (now **Equation 3**). It is now also defined in the caption of **Fig. 1c** where it first appears.

Page 4: The subscript, *app*, stands for approximation and $\Delta G_{f,app}(T)$ is defined in **Equation 3**.

-- For training their SISSO model, the authors used 262 compounds with 2,991 (T, G^Δ) chosen randomly. But the test set of 47 compounds contain only 558 (T, G^Δ). What is the difference between 2,991 and 558? It is not discussed in the paper.

The training and test set compounds were chosen randomly from the same pool of experimental data using a random number generator to identify which 15% of compounds would be reserved for testing. This is stated on page 5 of the manuscript – “A training set of 262 compounds with 2,988 (T, G^δ) points was randomly selected from 309 inorganic crystalline solid compounds with experimentally measured $G^\delta(T)$ (**Fig. 1a**) and was used for descriptor identification. The remaining 47 compounds with 558 (T, G^δ) points were reserved for testing.” This is now also included in the introduction.

Page 3: The descriptor is identified using experimental data²⁹ for 262 solid compounds and tested on a randomly chosen excluded set of 47 compounds with measured $G(T)$ and 131 compounds with first-principles computed¹⁶ $G(T)$.

-- It is unclear if the Volume (from high-throughput DFT) chosen by the authors also correspond to the same structure for which the experimental free energy is measured. They should add space group and the magnetic state to their Table S1 (at least for the experimental data from FactSage).

This is an excellent suggestion. We now include the Materials Project ID and optimized spacegroup in **TableS1**. Regarding magnetization, compounds with magnetic ions are initialized in a high-spin ferromagnetic configuration as described here:

https://materialsproject.org/docs/calculations#Total_Energy_Calculation_Details. Other computational details can be obtained from Materials Project with the ID now provided in **TableS1**.

-- Since the experimental free energy data span a wide temperature range, did the authors also take into account the changes in the crystal structure brought about by the structural or magnetic phase transitions in the considered temperature range? For example, take CaTiO₃. The authors use CaTiO₃ (from FactSage) in their training set and they give a value of 1500 for the T_{max} . It is known that CaTiO₃ undergoes structural phase transition as a function of temperature (Ali et al. Journal of Solid State Chemistry vol. 178, pages 2867-2872, year 2005). How did the authors account for phase transformations (in general) in their model? This is not discussed in the paper.

We agree that this should have been discussed in more detail. The 0 K ground-state structure reported in the Materials Project database (which includes the Inorganic Crystal Structure Database) was used to generate the *volume* (per atom) term used for training. This was done because one cannot know *a priori* for an arbitrary composition (which does not have measured thermochemical data), which structure is the experimentally realized ground-state.

FactSage provides phase transition temperatures for all compounds considered. For all compounds, we obtained $G(T)$ for the ground-state structure reported in FactSage at 298 K from $T = 298$ K until T_{max} (reported in **TableS1**), which is the maximum temperature before the first reported phase transition occurs (according to FactSage).

Regarding CaTiO₃, while we won't dispute the phase transitions in this material, FactSage reports the first solid-solid phase transition at 1530 K, hence motivating the inclusion of data up to 1500 K (T_{max}) for training/testing.

Please see the additions to the **Methods** section provided below.

Page 15: Compound data was extracted only at temperatures where the 298 K solid structure persists as reported in FactSage.

Page 17: **Structure considerations.** For training, we used 0 K ground-state structures (and magnetic configurations) reported in Materials Project. From this calculation result, we retrieved the volume (per atom) that is then used at all temperatures to generate $G^{\circ}(T)$ as shown in **Equation 4**. For a given composition, one could compute $G^{\circ}(T)$ for any number of structural or magnetic configurations and compare the $G(T)$ that results. For the purposes of training and testing, we consider only the calculated ground-state because this is likely the approach that would be used in practice for the application of the model to new materials which have available calculated but not experimental data.

-- A related question then is the following: In their 309 compounds from FactSage, how many have one or multiple structural or magnetic phase transitions within their reported T_{max} value? Do they have any statistic about this point?

As we hope the previous comment and related revisions clarify, there are *no* structural phase transitions reported by FactSage between 298 K and T_{max} as this fact is used to determine T_{max} in each case.

-- Is it problematic or a significant limitation of their approach if the training data does not describe or capture phase transformations? I can immediately see that it will be difficult to choose a volume if a compound undergoes phase transformation as a function of temperature. This should also affect the number of atoms in the unit cell and the phonons dispersion curves (QHA phonons).

For training and testing, we only used the ground-state structure. Our model does not preclude someone from generating $G^\delta(T)$ for a range of structures (and magnetic configurations). For instance, one could calculate the many structures of MnO_2 , obtain their formation enthalpies, $\Delta H_f(0 \text{ K})$, and volumes (normalized per atom in the calculated structure), and obtain $G^\delta(T)$ by **Equation 4** and $G(T)$ by **Equation 1** to identify which polymorph (and magnetic configuration) is the ground-state as a function of temperature. This is the approach used for the metastability analysis described in **Fig. 5-6** where every entry (ground-states and higher energy polymorphs) in the Materials Project database was considered and the ground-state was determined at each temperature and for each composition. Note that the number of atoms in the unit cell does not affect the model as the volume term is normalized per atom in the calculated structure. We've added a section describing the application of the descriptor to the Methods section to clarify these points.

Page 18: **Application of the descriptor.** To obtain the Gibbs formation energy for a given structure, one must first perform a DFT total energy minimization of the structure. From this, the atomic volume is determined as the volume of the calculated cell divided by the number of atoms in the calculated cell. G^δ can then be computed by **Equation 4**. Calculating the Gibbs energy, $G(T)$, using **Equation 1** requires the formation enthalpy, ΔH_f , calculated using DFT. If the analysis of interest concerns only one composition (chemical formula), then this is the final step and the relative energies of all structures with this composition can be compared using $G(T)$. If the analysis of interest considers various compositions (e.g., for convex hull stability or thermochemical reaction analysis), the elemental energies must be subtracted to obtain the Gibbs formation energy, $\Delta G_f(T)$ by **Equation 2**. Notably, ΔH_f and volumes calculated by DFT are tabulated for many thousands of structures and the elemental $G(T)$ are also tabulated for at least 83 elements. An important point is that users of the descriptor for $G^\delta(T)$ are free to generate ΔH_f and volumes for any number of structural or magnetic configurations for a given composition and compare how $G(T)$ might be sensitive to the changes in structure and magnetism.

-- It appears that the authors have taken the Volume data from the Materials Project database. From Table S1, I also find several transition metal oxides. Can the authors comment on the importance of magnetism in the choice of volume as a descriptor? This is an important point because the experimental Gibbs free energy for an antiferromagnetic compound should have the volume information for the same crystal structure and magnetic configurations from DFT. Otherwise, the mathematical model makes no physical sense. Can the authors clarify?

Because data was obtained from Materials Project, only high-spin ferromagnetic initializations are considered. However, the volume change (per atom) from ferromagnetic to antiferromagnetic structures is not expected to significantly impact the computed values of $G^\delta(T)$. As one typical example, the Materials Project ferromagnetic ground-state has a volume of 144.033 \AA^3 for a 12-atom cell and the NRELMatDB antiferromagnetic ground-state has a volume of 141.636 \AA^3 for a 12-atom cell. The difference between the two configurations is therefore $0.2 \text{ \AA}^3/\text{atom}$ which leads to a difference in $G^\delta(1000 \text{ K})$ of 2.6 meV/atom , well within the expected errors of the model, DFT, and experimental measurement. While one example is by no means conclusive, this demonstration is representative of this effect that leads to an order of magnitude lower error than that inherent in DFT and the experimental thermochemical data.

--In the SISSO method, the authors split the experimental training data into 262 and 47 for training and testing, respectively. The SISSO then identifies a functional form for predicting the Gibbs energy that contains volume, temperature and mass as descriptors. The authors then go on to justify the meaning behind this relationship, which is very interesting. My question is the following – if the authors randomly choose different sets of samples with the same 262 vs 47 split from using the same 309 experimental data (at least they should explore five sets of random sampling), would they get the same mathematical relationship every time (including the functional form)? I am curious to understand the robustness or sensitivity of their mathematical model to the training data. If they get different mathematical models with a different set of independent variables, what are the implications? How will the errors on test set and validation set (DFT-QHA Phonon database) behave or change every time for each of those random samples? In the absence of these assessments, I do not think their SISSO model building is complete.

The SISSO approach as described in **Methods** evaluates $\sim 3 \times 10^{10}$ three-dimensional models. As expected, from this immense set, many descriptors achieve comparable performance to the one shown in **Equation 4** and so we do not claim this is the only descriptor capable of predicting $G^\delta(T)$. However, this fact does not preclude one from obtaining physical meaning from the model that is found by SISSO and shown in **Equation 4** because this model is shown to be simple, intuitive, and comparably predictive of both training data and unseen (test) compounds.

Repeating the SISSO selection process many times is precluded because of the computational expense of doing so. For our problem, the SISSO selection process requires $\sim 1,500$ CPU hours. We note that none of the few papers that have employed SISSO have repeated the selection process for multiple train/test splits: <https://arxiv.org/abs/1710.03319>, <https://arxiv.org/abs/1801.07700>, <https://arxiv.org/abs/1805.10950>. In fact, it is uncommon in the field of machine learning to apply a learning algorithm to many splits of the training and testing data because doing so can cause “leaking” of the test data into the learning algorithm, defeating the purpose of the excluded test data set, which is intended to serve only to assess the predictive capabilities of the learning algorithm and resulting model. Repeating this process many times affords the opportunity to choose the model that happens to perform the best on the test data, which is recognized as not being a true indication of the predictive capabilities of the algorithm and model. Instead, we perform the train/test split a single time, ensuring that the SISSO selection process and fitting of **Equation 4** is truly blind to the test data. However, we appreciate the opportunity to clarify this point to the community, and at significant computational expense ($>18,000$ NREL supercomputer CPU hours), have performed the additional analysis to support this discussion, but we do not alter our original and “truly blind” model. With **Equation 4** found as described in the **Methods** section (i.e., resulting from a single train/test split) and not entertaining the idea of changing the model, we provide the following analysis to demonstrate that a) the performance of this model is insensitive to which compounds are considered training and which are considered testing and b) the features and functional forms in this model (**Equation 4**) are robust to changes of which compounds appear in the training and testing sets.

Page 16-17: **Descriptor sensitivity.** While the random splitting of the experimental set into training and test sets was performed only once, comparing the relevant properties for each set reveals that they are statistically similar, suggesting the model and SISSO process would yield similar results for an arbitrary random split of the experimental set (**Fig. S2**). To assess the robustness of the model on diverse training and test sets, we repeated the random split of the experimental set 1,000 times and evaluate the performance of **Equation 4** on each set. The MAD spans 37-42 meV/atom on the 85% training set and 26-54 meV/atom on the 15% test set, demonstrating that the reported 38 meV/atom for training and 46 meV/atom for testing (**Fig. 2**) are not outliers. As an added demonstration, the random split of the experimental set *and* subsequent SISSO selection process was repeated 12 times. In 10/12 runs, the descriptor shown in **Equation 4** appears in the top 3,000 of $\sim 3 \times 10^{10}$ models evaluated (top $\sim 0.00001\%$) in terms of root mean square deviation (RMSD) on the training set. Notably, there are many cases where very slight deviations of **Equation 4** also appear in the top models – e.g., replacing $\ln(T)$ with T or $T^{0.5}$. To validate the significance of the three features that comprise the descriptor – temperature, reduced mass, and atomic volume – we assess what fraction of the top 3,000 models contain these features for each of the 12 random train/test splits. Temperature is found to occur in 100% of the top models for each of the 12 random splits. Reduced mass and atomic volume each appear in $\sim 86\%$ of the top 3,000 models on average over the 12 random splits. This analysis was conducted on only the very best models (top $\sim 0.00001\%$) and reveals the significance of these three properties in predicting G^δ to be robust to the random split of the experimental data used to train and test the descriptor. Notably, the first term in **Equation 4**, $T \ln(V)$, appears as the feature with the highest correlation with G^δ in all of the 12 random train/test splits.

Page S2: **Figure S2.** Comparing the training and test sets (**Fig. 2**) in terms of the quantities relevant to the descriptor (**Equation 3**).

-- The section on the “effects of temperature and composition on stability”, where they are applying their training SISSO model to predict the Gibbs energy for ~30,000 compounds, is also problematic. The overlap between the 309 compounds for which the experimental Gibbs energy is known and the ~30,000 compounds is not clear. In other words, are crystal structure, chemical elements, Valence states of transition metal ions and their magnetic configurations etc, in both the 309 and 30,000 data samples identically distributed? It appears that the authors are extrapolating their results through learning from a small dataset of 309 compounds and applying it to a much larger set of 30,000 compounds. In the absence of experimental validation or uncertainty quantification, how can I trust their predictions?
 -- Under what circumstance does their SISSO model fail? Where can the predictions from the SISSO model be trusted? And more importantly, where does these models fail? If I cannot trust the model, what is the point of tabulating the Gibbs energy data for one million structures? There is no discussion about these crucial points.

We now address these important points within the Methods section.

Page 18: **Extension to new materials.** On the experimental training set of 262 compounds, the mean absolute deviation between experiment and the descriptor is 38 meV/atom (**Fig. 2**). This increases slightly to 46 meV/atom (**Fig. 2**) on the experimental test set and to 60 meV/atom on the computed (QHA) test set (**Fig. 3**). The residuals with respect to experiment are also mostly normally distributed, suggesting no systematic error in the model. The performance on the test set compounds is a demonstration of validated *prediction* accuracy or uncertainty on new predictions. These approximate “error bars” can be expected on *additional* new predictions to the extent that the sets used for training and testing are “comparable” to the new materials being predicted. The set we use for training and testing is quite diverse – 83 unique elements, binaries and multinaries, magnetic and nonmagnetic, metallic and insulating, etc. Additionally, the descriptor is relatively simple, having only four fit parameters (including the intercept) and three features (properties) that it depends upon. However, it has not been benchmarked for non-stoichiometric compounds or compounds with defects. For example, one could not expect to obtain the temperature-dependent defect formation energy using our descriptor because this was not benchmarked. Our model is also not capable of predicting the melting point of compounds. $G^\delta(T)$ is for the solid phase and can be obtained even well above a compound’s melting point, where the liquid phase has more negative Gibbs energy. We report substantial evidence that the descriptor *is* predictive over a wide range of stoichiometric solid compounds with a diverse set of chemical and physical properties.

-- The section on “Thermochemical reaction equilibria” is interesting and may have important implications. From Figure 4, it appears that their models are very accurate. A brief discussion on the limitations can help the readers.

We now conclude this section with a brief discussion of limitations.

Page 9: The accuracy of the descriptor-predicted reaction energies for new systems will be dependent not only on the effectiveness of $G^\delta_{SISSO}(T)$ to approximate $G^\delta_{exp}(T)$ but also on the extent to which DFT-predicted ΔH_f agrees with experiment as both parameters are required to obtain $\Delta G_f(T)$ (**Equation 2**) and therefore $\Delta G_{rxn}(T)$.

-- I would also like to bring to the attention of the authors a couple of recent works on machine learning

and materials discovery, where the Gibbs free energy data was not used for prediction of new compounds: Ren et al *Sci Adv* vol. 4, eaaq1556, year 2018 and Balachandran et al *Nat Commun* vol. 9, article number: 1668, year 2018.

We thank the reviewer for bringing these articles to our attention. Because Gibbs free energy data is not typically available or calculated, it is not surprising that there are cases where temperature-independent thermodynamics are sufficient to discover new materials (as has been shown by several of the authors of this manuscript). This raises an important point that we've now included in the **Discussion** section with reference to these papers.

Page 14: Therefore, when combining $G^\circ(T)$ with ΔH_f to determine the Gibbs formation energy, $\Delta G_f(T)$, errors in these approaches will be additive, emphasizing the need for new or beyond-DFT methods to calculate ΔH_f when extremely high accuracy is required for a given application. However, there are many examples where DFT-computed ΔH_f was used successfully to realize new materials⁵⁶⁻⁵⁸ and the incorporation of temperature effects using the SISSO-learned descriptor for $G^\circ(T)$ should only enhance these efforts.

Given these major concerns, I do not recommend this manuscript for publication in the Nature Communications journal.

We greatly appreciate the careful review which has considerably improved our manuscript, especially in terms of clarity in the presentation of the approach, and further validation of the robustness of the results and their implications. We hope that with these significant improvements and clarifications, the manuscript can now be recommended for publication.

Reviewer #2 (Remarks to the Author):

*The paper describes a new descriptor for estimation of the difference between the temperature dependent Gibbs free energy and the standard Helmholtz enthalpy. The descriptor is designed by the aid of machine learning techniques and represents a simple function of atomic volume, the reduced atomic mass and temperature. In general, it would be beneficial to have a simple and accurate way to estimate the thermodynamic parameters of solid compounds from basic properties such as volumes, masses and constituent elements. It is not a new idea and in the last at least 100 years there were several related studies published (e.g. Latimer, *J. Am. Chem. Soc.* 43, 818 (1921)). Reading carefully the submitted manuscript I am not convinced that this studies provide a "magic" formula that substantially improves the accuracy of these simple predictions and that could be use for precise thermodynamic considerations of solid compounds stabilities. I thus do not see significant advancement in the prediction of thermodynamic parameter of solid compounds that would grant rapid publication in Nature communication. Nevertheless, the studies are definitely of interest to thermodynamics community, but more appropriate for standard journal such as *Thermochimica Acta*. Below I give more details, justification and comments.*

We thank the reviewer for recognizing the relevance of our results to the thermodynamics community and noting that the problem we are addressing has been the subject of considerable investigation for at least 100 years.

We first clarify a few apparent misunderstandings. The descriptor predicts the difference between the Gibbs free energy and the standard formation enthalpy (not the Helmholtz energy) (**Equation 1**). The somewhat related studies (e.g., Latimer 1921) predict the standard entropy at 298 K, S_{298} , which does not enable the prediction of the temperature-dependent Gibbs energy or Gibbs formation energy as is possible with our descriptor. To do this, one also needs the temperature-dependence of S and of the enthalpy, H , which cannot be obtained from Latimer 1921 or Spencer 1998 but are included in our descriptor for $G^\circ(T)$. This is crucial because it is the temperature-dependent Gibbs energy that enables the calculation of stability and reaction energies as a function of temperature.

Detailed comments:

1) The title is a bit misleading as it suggest that a descriptor for prediction of the Gibbs free energy is given, while authors provide a formula for estimation of $G_{\text{Hf}}(298\text{K})$ only.

This might at first seem to be the case, but our descriptor enables the determination of $G(T)$ when the standard formation enthalpy is known. So, the descriptor could easily be reformulated to directly predict $G(T)$ by adding the standard formation enthalpy. That is, $G(T) = f(G^\delta, \Delta H_f)$ (**Equation 1**). ΔH_f is available for more than one million DFT-computed structures and we provide the critical link which enables $G(T)$ to be obtained trivially from ΔH_f .

2) Eq. 3: my understanding is that the terms besides DH_f should vanish at 298K, which is not true. These vanish at $T=324\text{K}$. The authors should check the correctness of the formula.

Thank you for checking this essential equation, but we've confirmed that the equation is correct. No terms in **Equation 3** (now **Equation 2**) are equal to zero at 298 K or 324 K. It is true that for elements, $H_i(298\text{ K}) = 0$, but $G_i(T) = H_i(T) - TS_i(T)$ and $S_i(T)$ does not equal 0 at 298 K or 324 K. Consider the following for Al_2O_3 from Barin's *Thermochemical Data of Pure Substances* (and also FactSage):

$$\Delta H_f(298\text{ K}) = -1656864.0\text{ J/mol} = -3.43\text{ eV/atom}$$

$$G_{\text{Al}}(298\text{ K}) = -8430.2\text{ J/mol} = -0.09\text{ eV/atom}$$

$$G_{\text{O}_2}(298\text{ K}) = -61131.9\text{ J/mol} = -0.32\text{ eV/atom}$$

Referring to **Equation 1** for $G^\delta_{\text{Al}_2\text{O}_3}$:

$$G^\delta(T) = G(T) - \Delta H_f(298\text{ K})$$

$$G_{\text{Al}_2\text{O}_3}(298\text{ K}) = -1672457.2\text{ J/mol} = -3.47\text{ eV/atom}$$

$$\text{Therefore, } G^\delta_{\text{Al}_2\text{O}_3}(298\text{ K}) = -15593.2\text{ J/mol} = -0.04\text{ eV/atom}$$

From **Equation 3** (now **Equation 2**):

$$\Delta G_{f,\text{Al}_2\text{O}_3}(298\text{ K}) = G_{\text{Al}_2\text{O}_3}(298\text{ K}) - 2G_{\text{Al}}(298\text{ K}) - 1.5G_{\text{O}_2}(298\text{ K})$$

$$\Delta G_f(298\text{ K}) = -1563898.8\text{ J/mol} = -3.24\text{ eV/atom}$$

Comparing with NIST JANAF, which reports $\Delta G_f(298\text{ K}) = -1563850\text{ J/mol} = -3.24\text{ eV/atom}$ for the 298 K gamma phase - <https://janaf.nist.gov/tables/Al-098.html>

Note 1: 324 K is not a temperature where thermochemical data is consistently provided in NIST JANAF, Barin's *Thermochemical Data of Pure Substances*, or FactSage.

Note 2: In the revised manuscript, **Equation 3** has become **Equation 2**.

3) Uncertainty of Eq. 5: According to Fig. 2 the uncertainty in Eq. 5 (descriptor) is as large as 0.1 eV/atom~10 KJ/atom. According to studies of Latimer, *J. Am. Chem. Soc.* 43, 818 (1921) or Spencer *Thermochimica Acta* 314, 1 (1998) the entropy contribution to the Gibbs free energy per atom is on average ~10 kJ/atom at 300K and 33 kJ/at at 1000K. The uncertainty of Eq. 5 is thus very large, which in

my opinion prevents quantitative estimates as claimed by the Authors. I have checked this on a few compounds reported in the Table submitted with the manuscript and the difference was indeed large.

Latimer [1921] only predicts S_{298} (the standard entropy at 298 K) and not $G(T)$ or $G^\delta(T)$. The magnitudes of S_{298} are indeed small. This can be seen also in our **Fig. 1a** which shows experimentally determined TS for 309 compounds. At 300 K, this quantity is only as large as ~ 0.2 eV/atom, or ~ 0.06 kJ/atom/K. Importantly, in addition to S_{298} one also needs $S(T)$ and $H(T)$ to obtain $G(T)$ or $G^\delta(T)$. The range of values for $G^\delta(T)$ is much larger, as shown in **Fig. 1a, 2, and 3**, with magnitudes as large as 1.5 eV/atom (150 kJ/atom). We now also report *relative errors* on the test set at 1000 K and 1800 K.

Page 6: Approximately 1/3 compounds considered have measured $G^\delta(1800\text{ K})$ and the mean absolute deviation (MAD) between G^δ_{SISSO} and G^δ_{exp} is found to increase from 53 meV/atom to 92 meV/atom from 1000 to 1800 K on the 47 compound test set. However, the mean absolute *relative error* actually decreases from 14% to 11% over this same range on the test set.

4) *Following comment 3, the Authors should provide a set of a few figures comparing predicted $G-H_f$ and the measured ones. Such experimental data exist for some compounds they consider.*

We believe the reviewer is referring to G^δ which is defined in **Equation 1** and shown already for every compound considered for training and testing the descriptor – **Fig. 2** compares experimentally measured and predicted G^δ for all 309 compounds from FactSage and **Fig. 3** compares calculated and predicted G^δ for all 168 compounds in PhononDB. In these figures, the subscript, *exp*, indicates experimental data, the subscript, *SISSO*, descriptor-predicted data, and the subscript, *QHA*, data computed by the quasiharmonic approximation. To clarify this crucial point, we have systematically ensured that each notation is consistently defined at the first instance of its use in the manuscript.

We value the reviewer's efforts in comparing our results to those in the existing literature and for recognizing the substantial efforts made for the last 100 years in predicting the thermodynamics of solid compounds. With the provided clarifications, we expect the reviewer will be convinced of the impact of our results as a significant advance on existing efforts in this area.

Reviewer #3 (Remarks to the Author):

This paper presents an application of a previously published ML method for symbolic regression (SISSO) to the prediction of the temperature dependent Gibbs energy of crystalline solids. The paper is sound and technically interesting and should be published. I have some suggestions for small improvements below.

There is no obvious innovation in the machine learning as this appears to be an essentially unchanged application of the previously published SISSO method. However, I do not know of other work applying ML to the temperature dependent Gibbs energy. Further, SISSO and other symbolic regression techniques are not commonly applied to materials science and expanding the scope of what ML for materials science means is in the broader interests of the community. In this way, I expect this paper to be of interest to scientists already interested in ML for materials science.

We appreciate the reviewer's positive comments regarding our manuscript and their recommendations for improving it.

I have two technical points I would like to see addressed.

Point 1

The results, such as those in Fig 4b, suggest a clear dependence of accuracy on temperature. A discussion of whether this is due to the model fitting or to extrapolation could be insightful. In particular, showing error plots such as in Fig 2 as a function of temperature would help answer the question: "Does

the model fit the high T data less well or is there something else going on?” but I’d like to see a discussion of this question in general.

We agree this is an interesting point. Please see the newly introduced text and figures.

Page 5-6: Notably, there is some T -dependence on the magnitude of residuals, with larger deviations as T (and therefore the magnitude of G^δ) increases (Fig. S1). There are three plausible reasons for this: 1) the magnitude of G^δ being predicted increases so at fixed relative error, the magnitude of the residuals is larger, 2) the number of compounds with measured $G^\delta(T)$ decreases as T increases, and 3) the physics dictating G^δ at high T are more complex due to e.g., significant anharmonic vibrational effects that are less accurately captured by the simple model of Equation 4. Approximately 1/3 of the compounds considered have measured $G^\delta(1800\text{ K})$ and the mean absolute deviation (MAD) between G^δ_{SISSO} and G^δ_{exp} is found to increase from 53 meV/atom to 92 meV/atom from 1000 to 1800 K on the 47 compound test set. However, the relative MAD actually decreases from 14% to 11% over this same range on the test set, supporting reason (1) as a primary driver for the increasing residuals at elevated temperature. Violin plots of residuals for the training and test sets as a function of temperature are shown in Fig. S1.

Page S2: Figure S1. Violin plot of residuals by temperature for the experimental dataset shown in Fig. 2. Above – training set; below – test set. Each violin is a kernel density estimate of the residuals at each temperature. Within each violin is a box-and-whisker plot which provides the mean residual as a white dot, the positive and negative quartiles as the wide shaded region (box) and the minimum and maximum residual as the ends of the thin line (whisker). The violins are scaled to have constant width at each temperature. The numbers at the top of each violin correspond with the number of compounds with data at each temperature. The retrieval of data at each temperature is described in “Data retrieval” in the Methods section.

Point 2

This paper does a poor job situating itself in the work of symbolic regression in general (though notably the original SISSO does not do a good job here either).

The authors compare to Eureka, but do not cite the most famous use of this [citations deleted for brevity].

We appreciate the context and useful articles which help better position the approach used in this work in the larger picture of symbolic regression for materials/physical science. We have revised the introduction to better place our work in the context of symbolic regression.

Page 2-3: Techniques based on symbolic regression have also shown that fundamental physics can be algorithmically obtained from experimental and computed data in the form of optimized analytical expressions of intrinsic properties (features).²⁵⁻²⁷ In this work, we apply a recently developed statistical learning approach, SISSO (sure independence screening and sparsifying operator)²⁸, to search a massive ($\sim 10^{10}$) space of mathematical expressions and identify a descriptor for experimentally obtained $G(T)$ that for the first time enables $\Delta G_f(T)$ to be readily obtained from high-throughput DFT calculations of a single structure (i.e., a single unit cell volume).

Minor points on the presentation:

* *Fig 1: I assume this is the data from FactSage and PhononDB that is used throughout the paper, but that is not obvious from the caption or the text.*

This is now described in the Fig. 1 caption.

Page 3: **Figure 1. a)** Experimentally obtained thermodynamic functions of 309 inorganic crystalline solid compounds obtained from FactSage. G° is defined in **Equation 1**. H° is the temperature-dependence of the enthalpy normalized to be zero at 298 K (**Equation S1**), S is the absolute entropy, and T is temperature. The subscript, *exp*, indicates the quantity is obtained from experimental data.

* *Fig 1b: I find this figure nearly impossible to read. The desaturated colors are hard to tell apart and the most desaturated ones are hard to see at all. I'm sure you can find a better way to display this data.*

We have modified **Fig. 1b** to make it easier to read. It is meant to show that the nature of the elements significantly affects the magnitude of $G_f(T)$ and highlight where G_N and G_C sit on this spectrum as they are referred to in the text. The Gibbs energies of all the elements are provided as a .json file in the linked github repository – github.com/CJBartel/predict-gibbs-energies.

* *Fig 1c: The error bars are not defined in the caption.*

Thank you for pointing this out. The caption now includes their definition.

Page 4: **c)** Mean absolute error in assuming a cancellation of solid vibrational entropy between the compound and the elements comprising it. $\Delta G_f(T)$ is defined in **Equation 3**. The subscript, *app*, stands for approximation and $\Delta G_{f,app}(T)$ is defined in **Equation 4**. The subscript, *exp*, indicates the quantity is obtained from experimental data. The error bars are standard errors of the sample mean.

We appreciate the reviewer's suggestions that significantly improve the manuscript in terms of the context of the approach and providing a more detailed exploration of our model and expect they will support publication now that we have addressed all their points of concern.

Reviewer:

Patrick Riley

Google Accelerated Science

pfr@google.com

Reviewers' comments:

Reviewer #1 (Remarks to the Author):

I appreciate the detailed response of the authors. I still have one question that I believe the authors' have not provided a convincing argument in the manuscript. This is regarding the calculation of $G(T)$ for compounds that undergo temperature-driven structural phase transitions AND that undergo magnetic transitions. The current total energy arguments are very generic and not helpful in deciphering the limitations.

Can the authors discuss specific examples in the manuscript (in section Methods or other parts where appropriate in authors' judgement), where their approach is able to predict the transitions correctly and where the transitions are not in agreement with experimental data (preferably from the ~29,000 prediction set)? In my opinion, this will provide a good benchmark for experimentalists to determine where to trust and where more work is needed (eg., lack of anharmonicity terms in the $G(T)$ expansion etc).

If the authors can provide a set of convincing examples, I would recommend this manuscript for publication in the Nature Communications journal.

Reviewer #3 (Remarks to the Author):

The authors have substantially addressed my concerns.

However, I have to disagree with several minor points in the rebuttal

Repeating the SISSO selection process many times is precluded because of the computational expense of doing so. For our problem, the SISSO selection process requires ~1,500 CPU hours. We note that none of the few papers that have employed SISSO have repeated the selection process for multiple train/test splits: <https://arxiv.org/abs/1710.03319>, <https://arxiv.org/abs/1801.07700>, <https://arxiv.org/abs/1805.10950>. In fact, it is uncommon in the field of machine learning to apply a learning algorithm to many splits of the training and testing data because doing so can cause "leaking" of the test data into the learning algorithm, defeating the purpose of the excluded test data set, which is intended to serve only to assess the predictive capabilities of the learning algorithm and resulting model.

First, it is absolutely common in ML practice to build multiple models on related sets of data to get a better estimate of generalization (commonly called cross-validation). It is true that you often will exclude a test set from the (as was done here) that is not included as training in any model.

Second, the claim of this being a large expense is not credible (unless there is some reason this calculation requires the supercomputing resources which it does not appear to me that it does). Standard cloud pricing for 1500 CPU hours is ~\$70 and can be as cheap as ~\$15
<https://arxiv.org/abs/1805.10950>
<https://arxiv.org/abs/1805.10950>

Overall, I am supportive of publication from the perspective on an ML expert.

Note for all reviewers: We very much appreciate your helpful comments and suggestions, which has helped us address remaining points of confusion in the manuscript. Reviewer comments are shown in *italic black text*, with our responses shown in blue. In the revised manuscript, changes are highlighted in **yellow** and also included in the response to the comment that motivated the change.

Reviewers' comments:

Reviewer #1 (Remarks to the Author):

I appreciate the detailed response of the authors. I still have one question that I believe the authors' have not provided a convincing argument in the manuscript. This is regarding the calculation of $G(T)$ for compounds that undergo temperature-driven structural phase transitions AND that undergo magnetic transitions. The current total energy arguments are very generic and not helpful in deciphering the limitations.

Can the authors discuss specific examples in the manuscript (in section Methods or other parts where appropriate in authors' judgement), where their approach is able to predict the transitions correctly and where the transitions are not in agreement with experimental data (preferably from the ~29,000 prediction set)? In my opinion, this will provide a good benchmark for experimentalists to determine where to trust and where more work is needed (eg., lack of anharmonicity terms in the $G(T)$ expansion etc).

If the authors can provide a set of convincing examples, I would recommend this manuscript for publication in the Nature Communications journal.

We're grateful to the reviewer for probing this important question and showing that the existing manuscript is unclear in this regard. This problem, while undoubtedly interesting, is beyond the scope of our descriptor for a few reasons: 1) the energy differences between magnetic or structural configurations (i.e., polymorphs) that change as a function of temperature are well within the magnitude of errors for our descriptor (~40 meV/atom on average), 2) the descriptor was trained on compounds that do not exhibit phase transitions over the temperature range where data was collected for training (as described in **Methods – Data retrieval**), 3) at fixed composition, the only parameter that can vary as magnetic or structural configuration is varied is V and the functional form of G^δ dictates that as V increases, G^δ decreases at all temperatures, thus the case where a denser lattice becomes more stable than a less dense lattice cannot occur using our descriptor. Therefore, we do not agree that providing a set of examples where our descriptor is applied to polymorphic or magnetic phase transitions is appropriate for or illustrative of the general problem at hand, even in cases where our descriptor would predict these transitions correctly. As an alternative that we believe you will also deem acceptable, we now provide the explicit and direct guidance regarding the prediction of polymorphic (structural or magnetic) phase transitions using the descriptor identified in this work, and that it should not be employed for these purposes.

The agreement between theory and experiment for polymorph energy ordering and phase transition temperatures is a grand challenge even for first-principles (and experimental) approaches that is often unable to be resolved even using state-of-the-art, high-fidelity, computationally expensive approaches – e.g., see TiO₂ with Quantum Monte Carlo and lattice dynamics: <https://doi.org/10.1088/1367-2630/18/11/113049>. To address this in a high-throughput way is currently intractable and would require a devoted effort to this problem.

We emphasize where our descriptor has significant utility, and how we apply it in this work, is in the prediction of thermochemical stability of *compounds* relative to one another (i.e., stability over variable composition) and for the prediction of thermochemical reaction equilibria. The Gibbs energies that

determine both of these applications are only affected in a small way by changes in magnetic or structural configurations that might occur with temperature.

We've added text to this effect in the **Methods** and **Main text** which is copied here for your convenience:

Page 7: Importantly, V is the only structural parameter in **Equation 4** and therefore, at fixed composition (chemical formula), G^δ varies between structures (i.e., polymorphs) only as V varies and $G^\delta(V)$ dictates that less dense structures of the same composition will have more negative G^δ . Therefore, the prediction of polymorphic phase transitions is beyond the scope of this descriptor.

Page 18: As alluded to in the main text, the extension of the descriptor to correctly predict polymorphic phase transitions or temperature-driven magnetic transitions is not practical because the descriptor depends only on the mass, density, and temperature and the magnitude of the energy change for these transitions is typically smaller than the expected error bars of the descriptor. We report substantial evidence that the descriptor is predictive for stability of compounds relative to one another and for the prediction of thermochemical reaction equilibria over a wide range of stoichiometric solid compounds with a diverse set of chemical and physical properties.

Reviewer #3 (Remarks to the Author):

The authors have substantially addressed my concerns.

However, I have to disagree with several minor points in the rebuttal

Repeating the SISSO selection process many times is precluded because of the computational expense of doing so. For our problem, the SISSO selection process requires ~1,500 CPU hours. We note that none of the few papers that have employed SISSO have repeated the selection process for multiple train/test splits: <https://arxiv.org/abs/1710.03319>, <https://arxiv.org/abs/1801.07700>, <https://arxiv.org/abs/1805.10950>. In fact, it is uncommon in the field of machine learning to apply a learning algorithm to many splits of the training and testing data because doing so can cause "leaking" of the test data into the learning algorithm, defeating the purpose of the excluded test data set, which is intended to serve only to assess the predictive capabilities of the learning algorithm and resulting model.

First, it is absolutely common in ML practice to build multiple models on related sets of data to get a better estimate of generalization (commonly called cross-validation). It is true that you often will exclude a test set from the (as was done here) that is not included as training in any model.

Second, the claim of this being a large expense is not credible (unless there is some reason this calculation requires the supercomputing resources which it does not appear to me that it does). Standard cloud pricing for 1500 CPU hours is ~\$70 and can be as cheap as ~\$15

Overall, I am supportive of publication from the perspective on an ML expert.

We appreciate your support for publication. Your comments are all fair points and we have ensured that these issues are not raised or disputed in our revised manuscript. We're grateful for your reviews that have improved the manuscript and appreciate the suggestions for the ML aspects of the work based on your ML expertise.